# Maturation and substrate processing topography of the *Plasmodium falciparum* invasion/egress protease plasmepsin X

Sumit Mukherjee[1], Suong Nguyen[1], Eashan Sharma[1] & Daniel E. Goldberg [1] ✉

The malaria parasite *Plasmodium* invades a host erythrocyte, multiplies within a parasitophorous vacuole (PV) and then ruptures the PV and erythrocyte membranes in a process known as egress. Both egress and invasion are controlled by effector proteins discharged from specialized secretory organelles. The aspartic protease plasmepsin X (PM X) regulates activity for many of these effectors, but it is unclear how PM X accesses its diverse substrates that reside in different organelles. PM X also autoprocesses to generate different isoforms. The function of this processing is not understood. We have mapped the self-cleavage sites and have constructed parasites with cleavage site mutations. Surprisingly, a quadruple mutant that remains full-length retains in vitro activity, is trafficked normally, and supports normal egress, invasion and parasite growth. The N-terminal half of the prodomain stays bound to the catalytic domain even after processing and is required for proper intracellular trafficking of PM X. We find that this enzyme cleaves microneme and exoneme substrates before discharge, while the rhoptry substrates that are dependent on PM X activity are cleaved after exoneme discharge into the PV. The data give insight into the temporal, spatial and biochemical control of this unusual but important aspartic protease.

Malaria is caused by the parasitic protozoan *Plasmodium* and accounts for more than 600,000 deaths annually[1]. The majority of deaths are due to infection with *Plasmodium falciparum*. The disease symptoms result exclusively from the blood stage of infection during which the parasite replicates asexually within the host red blood cells (RBCs) by a process termed schizogony. Each schizont matures into 8–32 merozoites and, by rupturing the parasitophorous vacuolar membrane (PVM) surrounding the parasite as well as the red blood cell membrane (RBCM), exits from the host cells by a process called "egress"[2]. The egressed merozoites subsequently invade fresh RBCs. Given the importance of the intraerythrocytic cycles of egress and invasion in the spread of the infection, it is, therefore, of considerable interest to target these processes for the development of disease intervention strategies.

The *Plasmodium* aspartic protease plasmepsin X (PM X) is essential for replication of *P. falciparum* during the intraerythrocytic

cycle[3–5]. In mature schizonts, PM X is located in the exoneme secretory organelle and is involved in the proteolytic activation of the other exoneme-localized protease subtilisin-like protease 1 (SUB1). Upon release into the parasitophorous vacuole (PV), SUB1 encounters and activates downstream effectors such as serine repeat antigens (SERAs) and the merozoite surface proteins (MSPs)[6–8]. Concerted activities of these molecules lead to rupture of both the PVM and the RBCM, releasing the daughter parasites into the bloodstream. Importantly, PM X is druggable, as multiple classes of compounds targeting PM X have high efficacies both in vitro and in vivo as well as being active on different life stages[3–5].

Recent data based on PM X inhibitors and peptide substrate cleavage analysis, suggest that aside from SUB1, PM X has additional substrates in other secretory organelles that function during parasite invasion[5]. Among these substrates, the apical membrane antigen1

[1]Division of Infectious Diseases, Department of Medicine, and Department of Molecular Microbiology, Washington University School of Medicine, St. Louis, MO, USA. ✉e-mail: dgoldberg@wustl.edu

(AMA1), subtilisin 2 (SUB2) and the erythrocyte binding antigens (EBAs) are targeted to the micronemes[9,10], whereas the reticulocyte binding protein homologs (Rhs) are trafficked to the rhoptries[11]. These substrates are therefore spatially segregated from PM X in mature schizonts. How and where PM X gains access to these proteins for processing is a mystery.

The primary *Pfpm x* gene product (p64) consists of an N-terminal prodomain (PD) followed by a catalytic domain[4,12]. This precursor PM X is converted into multiple shorter forms (p52, p49 and p42) through autoproteolytic cleavage[5]. The processing of PM X never goes to completion, as both unprocessed and processed forms are detected in terminal schizonts[3]. However, the significance of this partial processing remains unknown. Recent data demonstrate a PM X-specific inhibitor can bind to multiple processed forms of PM X, suggesting that they may be active[5].

In this work, we sought to address the following important question: how does the processing regulate PM X trafficking, activity, and substrate specificity, given the multiple processed forms of PM X and the differentially localized substrate repertoire? We find that PM X has unusual characteristics for an aspartic protease and provide cellular evidence to explain its ability to process substrates in different compartments.

## Results

### PM X undergoes autocleavage at multiple sites that are independent of each other

Previously we reported that multiple processed and unprocessed forms of PM X remain in terminal schizonts[3]. To shed light on the significance of PM X processing, it was of interest to determine the cleavage sites. However, because of low abundance of PM X in parasites, we were unable to extract enough native enzyme from parasite lysates for N-terminal sequence analysis. We therefore took advantage of the recombinant PM X (rPM X) that we could successfully express as a C-terminally 8x-histidine-tagged secreted protein in mammalian cells. The protein was a mixture of differentially processed polypeptides (Fig. 1a, b). Of these, one polypeptide of ~12 kDa molecular weight (p12) was detected only in the Coomassie blot but not in the western blot with an anti-His antibody, suggesting that it is an N-terminal fragment (red line and red arrow in Fig. 1a, b). To precisely determine the composition of p12, we excised the band from the gel and subjected it to proteomic analysis (Supplementary Data 1). This confirmed that p12 represents the cleaved N-terminal fragment from the PD of rPM X (Fig. 1a, red arrow). Immunoprecipitation with anti-HA antibody could pull down the differentially processed His-tagged polypeptides from an N-terminally HA tagged version of rPM X (Supplementary Fig. 1). This suggests that p12 stays bound to each of the processed forms. Among the higher molecular weight fragments that retained the C-terminal 8x-His tag, we were able to determine the sequence of the 42 kDa polypeptide (p42) by LC/MS (Fig.1 a, b, light blue arrows; Supplementary Data 2). Importantly, the N-terminus of this polypeptide ended at a nontryptic leucine residue (open arrow end on light blue line in Fig. 1a).

The amino acid sequence (IALE) that spans the N-terminal end of p42 conforms to the PM X substrate cleavage specificity defined by Favuzza et al.[5]. To test this further, we mutated the last two residues to alanine (IALE to IAAA, Supplementary Fig. 2). Such di-mutation had been shown to abolish cleavage of a fluorogenic peptide by isolated PM X[4,5]. This mutated version of PM X (rIAAA) expressed in mammalian cells, exhibited processing indistinguishable from that of wild type (Fig. 1c). 7 amino acids upstream of this cleavage site is another sequence motif (SSIE) that corresponds to the PM X peptide cleavage specificity (Fig. 1a, Supplementary Fig. 2). While mutation of this alternate site alone (SSIE to SSAA) had no effect, mutation of both sites together (rSS/IA^Dbl mut) prevented processing to p42 (Fig. 1c, d).

Importantly, the rSS/IA^Dbl mut still underwent normal cleavage at an upstream site, as shown by the presence of p12 in the Coomassie blot (Fig. 1d, left panel).

The p12 fragment (red arrows in Fig. 1a, b) ended at a nontryptic phenylalanine at the C terminus (Supplementary Data 1). To see if this fragment results from autoproteolytic cleavage, we mutated the amino acid sequence spanning the C terminal end (NFLD to NFAA, Supplementary Fig. 2). The resultant mutant (rNFAA) was still cleaved, although differently from that of the WT (Fig. 1c, d, gray arrow). LC/MS confirmed that this fragment was cleaved after an aspartate, which is 20 amino acids downstream from that of the WT derived fragment (Supplementary Data 3, gray arrow in Fig. 1a). The sequence flanking this new C terminal end (SDIQ) also conformed to the PM X substrate cleavage specificity. Simultaneous mutations of all four identified sites (rQuad^mut, Supplementary Fig. 2) completely blocked the processing of rPM X, resulting into the accumulation of the full-length form (Fig. 1c, d, right panels).

To further analyze these cleavages, we moved to cultured parasites and expressed PM X mutants as GFP-tagged second copies. For WT, we observed four bands (90, 77, 75 and 68 kDa, corresponding to the GFP fusions to p64, p52, p49 and p42, respectively). Mutation of NFLD and SDIQ together abolished processing to the 77 and 75 kDa but not to the 68 kDa form (Fig. 1e). Mutation of IALE and SSIE together blocked processing to the 68 kDa form, but had no impact on processing to the 77 or 75 kDa forms. Finally, a quadruple mutant (Quad^mut) resulted in accumulation of the full-length form. The data suggest that similar to the mammalian rPM X, PM X expressed in parasites undergoes autocatalytic processing at multiple sites that are independent of each other.

### Proteolytic processing is not necessary for parasite growth or for activity of PM X

Our second copy mutants were made in an endogenous PM X TetR-regulated knockdown line, PM X^apt (see ref. 3). Wild-type second copy PM X but not a catalytically dead mutant (D266G) can rescue the growth defect that is seen when parasites are grown in the absence of anhydrotetracycline (aTc) to deplete endogenous PM X (see ref. 3 and Supplementary Fig. 3). Surprisingly, each of the cleavage site mutants including the quadruple mutant (Quad^mut) was able to rescue growth in the absence of aTc (Fig. 2a). We analyzed the processing of PM X substrate proteins SUB1 and AMA1 by the different cleavage mutant forms of PM X when endogenous PM X was knocked down. As previously reported[3], withdrawal of aTc in the parental PM X^apt line results in a processing defect in conversion of the SUB1 54 kDa intermediate to the mature 47 kDa form (Fig. 2b). This PM X knockdown also impairs conversion of the AMA1 70 kDa precursor to the 60 kDa form. Processing of both proteins was fully restored by second copy expression of the different PM X cleavage mutants.

The observation that even the quadruple mutant could efficiently cleave SUB1 and fully restore parasite growth in the absence of endogenous PM X was of particular interest. To assess the enzyme activity of the second copy PM X, we pulled down WT and Quad^mut PM X from parasites using anti-GFP antibodies followed by in vitro substrate cleavage assays (Fig. 3 and Supplementary Fig. 4). As a control, we similarly pulled down the active site D266G PM X mutant from parasites. The Quad^mut PM X showed robust substrate cleavage activity comparable to that of the WT at pH 5.5 (Fig. 3). For both samples the activity dropped at pH 6.5 though there was a difference in residual activity between the Quad^mut and the WT enzyme. At pH 7.5, both samples lost their activity. As expected, the PM X inhibitor, CWHM-117 was able to block activity for both samples and the D266G did not show significant activity even at pH 5.5. Together, these data suggest that processing of PM X is not important for its function in parasites.

## In vitro activity of rPM X is correlated with acid-induced structural changes

For many proteases, the N-terminal prodomain (PD) remains tightly bound to the catalytic domain, acting as an endogenous inhibitor for the mature enzyme[13,14]. Removal of the PD is therefore necessary for enzyme activation. Surprisingly we noticed that the N-terminal fragment of the proenzyme remains noncovalently attached to the processed rPM X after isolation (Fig. 1b). To understand this, we carried out size-exclusion chromatography on WT rPM X that had been pre-incubated in different pH conditions (pH 7.5 and 5.5). Upon gel filtration under acidic conditions, rPM X eluted in a single peak that is identical to the one obtained at pH 7.5 (Fig. 4a, top panel, Supplementary Fig. 5). In contrast, hemoglobin tetramers, which dissociate to monomers under acidic conditions[15], demonstrated a shift in the elution profile at pH 5.5 (Fig. 4a, bottom panel).

For PM X an additional peak that corresponds to p12 was observed when a denaturing agent, guanidine hydrochloride (GuHCl) was added in the acidic medium (Fig. 4b, c). The p12 polypeptide remained associated with the main peak at pH 5.5 without denaturant (Fig. 4c) and was evenly distributed throughout the peak by SDS-PAGE and by dynamic light scattering (Supplementary Fig. 5). Together these data suggest that the activation of PM X does not involve complete removal of the N-terminal part of the PD. Consistent with this, the rQuad^mut PM X demonstrated robust substrate cleavage activity in acidic buffer (Supplementary Fig. 6). To further investigate PM X's dependency on acidic pH for activation, we examined the WT rPM X by circular dichroism (CD) spectroscopy. As indicated by changes in molar ellipticity, a significant decrease was observed in the alpha helical composition of rPM X at pH 5.5 as compared to pH 7.5 (Fig. 4d and Supplementary Table 1).

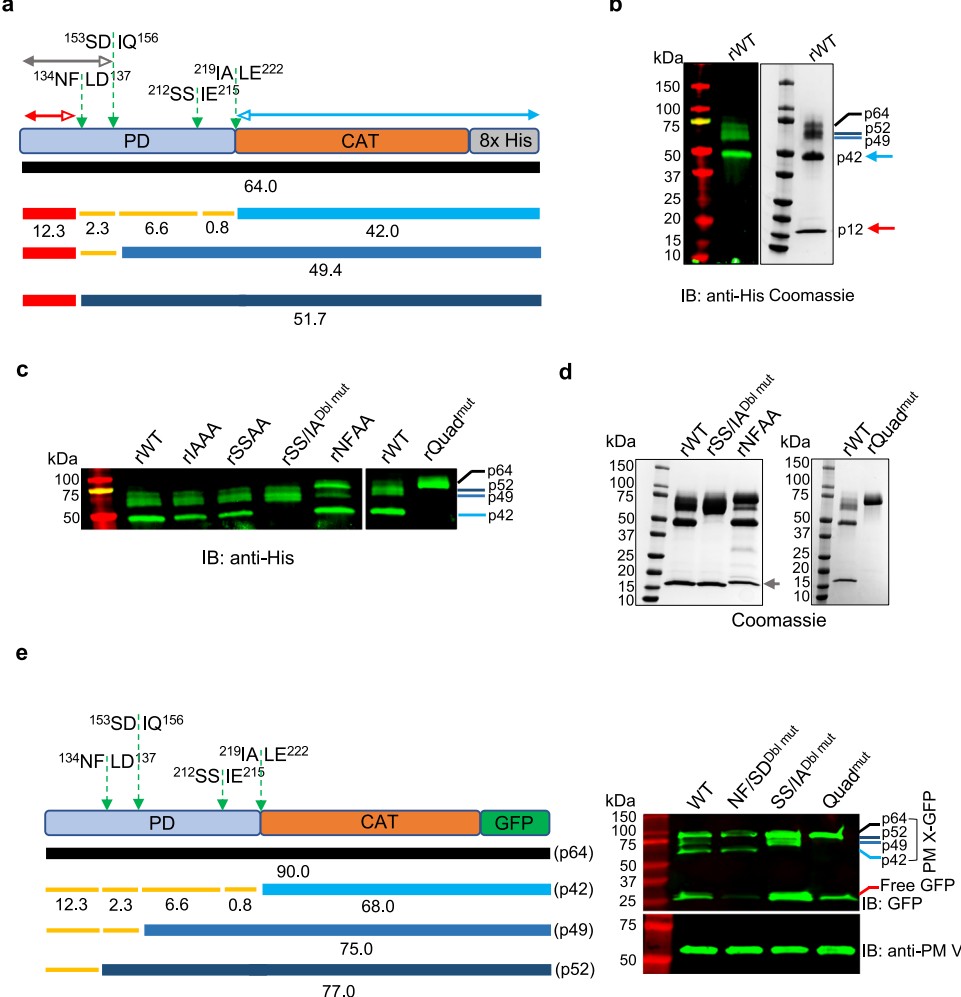

**Fig. 1 | Autoprocessing sites are conserved between the heterologously expressed and parasite PM X. a** Schematic of the likely identities of the different processed forms of C-terminally 8x His tagged PM X (rPM X) expressed from HEK293 cells. Predicted mass of the polypeptides based on their amino acid compositions are indicated. Thick lines indicate the processed fragments that were detected either by western blot or by Coomassie blot. Orange thin lines represent processed bands not detected. Amino acid sequence flanking the PM X autocleavage sites are indicated with positional information. Green dotted arrows represent the scissile bonds. PD prodomain, CAT catalytic domain. The red, blue and gray arrows represent the peptide coverage regions obtained from LC/MS. Filled arrow heads: tryptic ends, empty arrow heads: nontryptic ends. Colors correspond to the bands highlighted by same color arrows in **b** and **d**. Additional mutated constructs used in this study are depicted in Supplementary Fig. 1. **b** Left,

anti-His immunoblot; right, Coomassie stain of wild type rPM X (rWT). Anti-His immunoblot (**c**) and Coomassie gel (**d**) showing processing of rWT or mutant rPM X at multiple autocleavage sites. rSS/IA^Dbl mut : SSAA/IAAA double mutant; rQuad^mut : quadruple mutant with additional NFAA and SDAA mutations. **e** Expression of second-copy PM X in parasites. Left, schematic of the second copy PM X that was introduced into parasites as a C-terminally GFP-tagged construct. Labels as in **a**. Right, top panel: anti-GFP immunoblot, the assigned unprocessed and differentially processed forms indicated by color coded lines. Bottom panel: lysates from the same samples but blotted with anti-PM V antibody as loading control. NF/SD^Dbl mut : NFAA/SDAA double mutant; SS/IA^Dbl mut : SSAA/IAAA double mutant; Quad^mut : quadruple mutant with all four cleavage sites changed to AA in the P1' and P2' positions. Each experiment (**b**–**e**) was repeated at least three times and shown are representative blots. Source data are provided as a Source Data file.

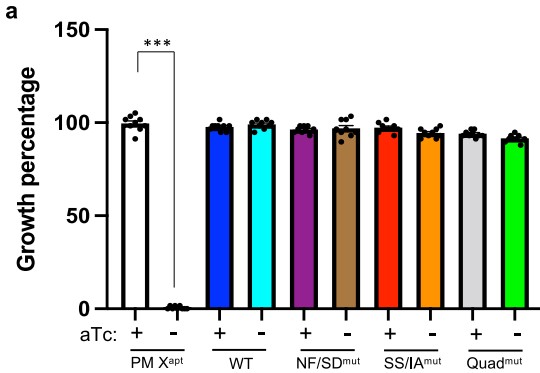

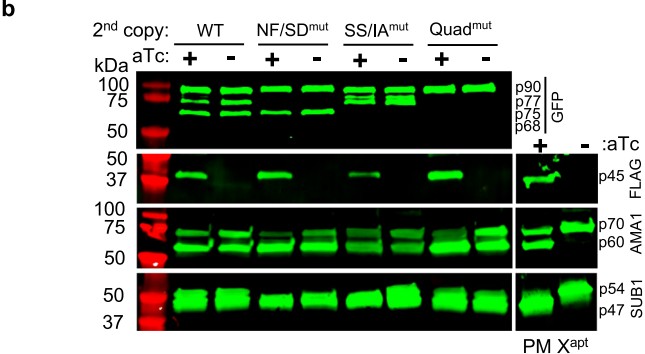

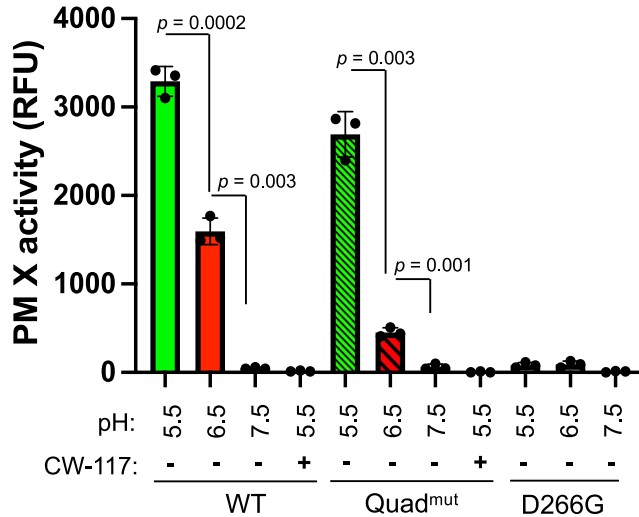

**Fig. 3 | Proteolytic processing is not necessary for in vitro activity of PM X.** Second copy PM X-GFP was purified from synchronized 42–45 h schizonts using anti-GFP antibody. The pulled down proteins were then incubated with fluorogenic Rh2N substrate peptide (1 μM) at the indicated pH. In control wells, CWHM-117 (1 μM) was added to inhibit PM X activity. Reactions were carried out for 1 h at 37 °C. Mean values from three independent experiments are shown and error bars represent standard deviations. Data were analyzed statistically by two-tailed Student's *t* test, *p* values are shown on the graph. Source data are provided as a Source Data file.

**Fig. 2 | Autocatalytic processing of PM X is not required for parasite replication and substrate cleavage. a** The parental PM X$^{apt}$ parasites or PM X$^{apt}$ parasites expressing the indicated PM X as a second copy were grown either in the presence or absence of aTc for 96 h. The starting parasitemia was 1% across different samples. Shown are the final parasitemia percentages after 2 erythrocytic cycles, normalized to that of the PM X$^{apt}$ parasites grown in the presence of aTc for 96 h. Mean values from three independent experiments are shown and error bars represent standard deviations. Data were analyzed statistically by two-tailed Student's *t* test. ***$p = 0.00002$. **b** The parasite lines from **a** were MACS synchronized for 3 h and were grown for the next 45 h either in the presence or absence of aTc. Parasites were then harvested and whole cell lysates were prepared. Western blots were performed to detect the expression of the second copy PM X-GFP (top panel), endogenous PM X-FLAG (second panel), AMA1 (third panel) and SUB1 (bottom panel). On the right is the PM X$^{apt}$ line without second-copy PM X expression. The antibodies used and the specific molecular weights of the different processed forms of proteins are indicated. This experiment was repeated two times and shown is a representative blot. Source data are provided as a Source Data file.

## The N-terminal PD piece is required for intracellular trafficking and functionality of PM X

To see how the PD regulates PM X function, we made truncation mutations in the PD and expressed the truncated PM X as C-terminally GFP-tagged second copies in a SUB1-3xHA parasite line (Fig. 5a). Specifically, we either deleted the sequence between the N-terminal signal peptide (SP) and the NFLD autocleavage site (delA) or the sequence between the NFLD and the downstream IALE cleavage site (delB). As a control, we made a third construct that lacked the entire PD (delPD). IFAs with mature schizonts showed that, similar to the WT PM X, the delB but not the delA or delPD was substantially colocalized with the SUB1-3xHA in exonemes (Fig. 5b, c, Supplementary Fig. 7). We believe that partial cleavage of GFP from the PM X during secretion (Fig. 1e) accounts for the imperfect colocalization of WT and delB PM X with SUB1. The delA and delPD, on the other hand, showed near complete colocalization with the ER resident protease PM V[16–18]. To further assess intracellular trafficking by an orthologous measure, we looked at the secretion of second copy PM X into the culture

supernatant from synchronized 41–44 h schizonts that were pre-treated with zaprinast. Zaprinast is a phosphodiesterase (PDE) inhibitor and causes premature egress of parasites by promoting discharge of effectors from apical secretory organelles in a protein kinase G (PKG)-dependent manner[19]. Western blots from fractionated samples showed that, similar to the endogenous PM X (anti-FLAG blot), both the WT and delB but not delA (anti-GFP blot) were partially secreted into the culture supernatant following a 45 min zaprinast treatment (Fig. 5d). Together with the IFAs, this data therefore suggests that the N-terminal part of the PD carries information that is critical for folding and trafficking of PM X beyond the ER.

In addition to trafficking, we also looked at the growth of the transgenic lines that were made in the PM X$^{apt}$ background. As shown in Fig. 5e, in the absence of aTc, the growth defect observed in the parental line could be restored in the WT or delB but not the delA-expressing parasite lines. We pulled down the different second copy PM X versions from parasite lines using anti-GFP antibodies and tested their activities on fluorogenic peptide. As shown in supplementary Fig. 8, the delB but not the delA PM X was active at pH 5.5. Therefore, the N-terminal fragment of the PD appears to be critical for PM X activity, likely by aiding correct folding of the protease.

## Autocatalytic processing of PM X occurs in a late secretory compartment

To understand the maturation of PM X further, we treated synchronized 40–43 h parasites expressing PM X-3xHA for 8 h with the cysteine protease inhibitor E64d to prevent egress but still allow organellar discharge as well as PVM rupture and RBCM poration[20,21]. Secreted proteins (supernatants) were then separated from those that remained intercellular by centrifugation. As shown in Fig. 6, similar to the vehicle (DMSO)-treated samples, both unprocessed and processed forms of PM X were detected in the culture supernatant of the E64d-treated samples. As expected, treatment with the protein kinase G (PKG) inhibitor compound 1 (C1), which blocks apical organellar

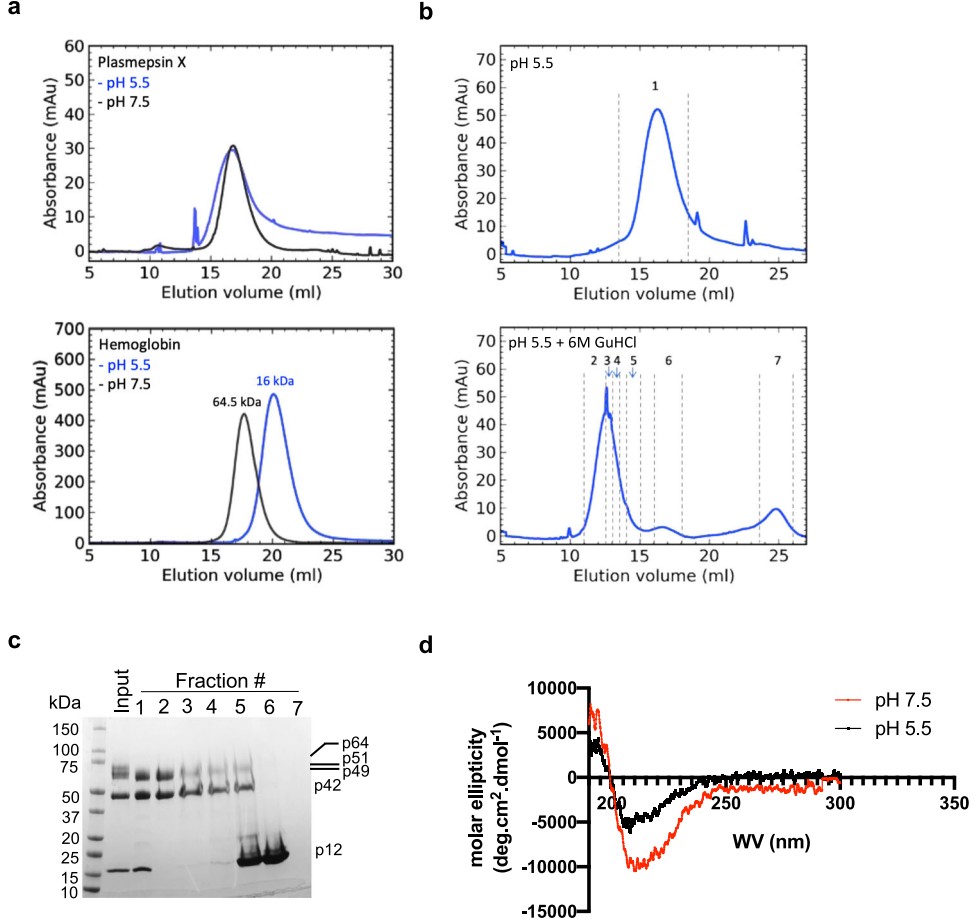

**Fig. 4 | The p12 polypeptide remains attached to rPM X under activation conditions. a** Overlay of the elution profiles of WT rPM X (top) or hemoglobin a (bottom) following size exclusion chromatography at the indicated pH. **b** Elution profile of rPM X following size exclusion chromatography at pH 5.5 in the absence (top) or presence (bottom) of 6 M guanidine HCl (GuHCl) as a denaturant. Samples were pretreated in the same buffer before loading onto the column. Numbers within dotted lines indicate the fractions that were collected to run on SDS-PAGE followed by Coomassie staining (**c**). Subfractionation analysis of the pH 5.5 peak is shown in Supplementary Fig 5. **d** Comparison of the molar ellipticity of WT rPM X preincubated at the indicated pH and analyzed by circular dichroism spectroscopy. Experiment was repeated three times in triplicate. Shown are the mean values from one representative experiment. Source data are provided as a Source Data file.

discharge in *P. falciparum*, blocked the secretion of PM X into the culture supernatant[19]. Importantly, the fungal metabolite brefeldin A (BFA), which prevents transport of newly synthesized secretory proteins out of the ER[22,23], completely blocked PM X processing (Supplementary Fig. 9). This indicates that all the processing steps happen in a post-ER compartment.

## PM X cleaves microneme and rhoptry substrate at different cellular locations

The processing defect of the micronemal protein AMA1 seen upon knockdown of PM X (Fig. 2b) provides a genetic confirmation for the previously observed inhibition of AMA1 cleavage by a PM X-specific inhibitor[4,5]. The same work also identified a number of rhoptry-localized proteins as substrates of PM X. Given that PM X was originally found to be colocalized with SUB1 in the exoneme secretory organelles, these observations suggest that the interaction of PM X with its substrates may be spatially regulated. We analyzed the processing of PM X substrates under conditions where organellar discharge was blocked by treatment with the C1. Control parasites were treated with E64d to block parasite egress without impairing organellar discharge. Processing of AMA1 (microneme), SUB1(exoneme) and Rh5 (rhoptry) was assessed. As shown by western blot, both AMA1 and SUB1 were processed in cells treated with C1 (Fig. 7a). On the other hand, treatment with C1 blocked the processing of Rh5, resulting in the accumulation of the 60 kDa precursor form (Fig. 7b). For Rh5 we could only

detect the presence of the processed 50 kDa form when parasites were allowed to discharge the apical organellar contents. As expected, all three proteins were accumulated in their precursor forms when samples were cultured in absence of aTc to knock down PM X. Finally, as a control for organellar discharge, we analyzed the PV-localized SUB1 substrate SERA5. Processing of its 126 kDa precursor to lower molecular weight forms was detected only when activated SUB1 was discharged into the PV (+aTc/+ E64d sample). Additionally, AMA1 was found to be correctly localized in terminal schizonts even in the absence of aTc, suggesting that the PM X-mediated cleavage is not necessary for trafficking to the micronemes (Supplementary Fig. 10). Altogether, the data suggest cleavage of different substrates by PM X is regulated spatially in *P. falciparum*.

Importantly, the fact that PM X could cleave the microneme-localized substrate AMA1 in the presence of C1 suggested that the two proteins encounter each other intracellularly before organellar discharge. To further understand how PM X interacts with non-exoneme proteins, we carried out high resolution 3D-confocal Airyscan microscopy with C1-treated terminal schizonts expressing PM X-3xHA or SUB1-3xHA. In cells that were dual-labeled with anti-AMA1 and anti-HA antibodies, we observed differences in the punctate patterns of expression of AMA1 compared to PM X or SUB1, but there were regions that showed complete overlap between the protein pairs (Fig. 8a, b, 2D and 3D snapshots respectively). This is of particular interest since an earlier study reported that AMA1 and SUB1 are targeted to different

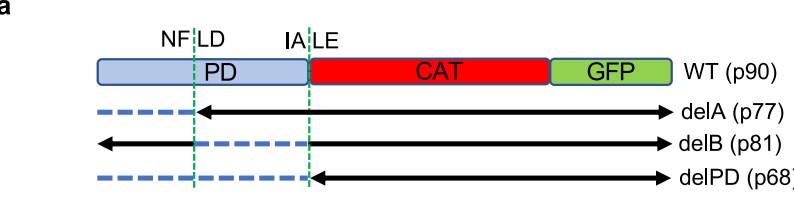

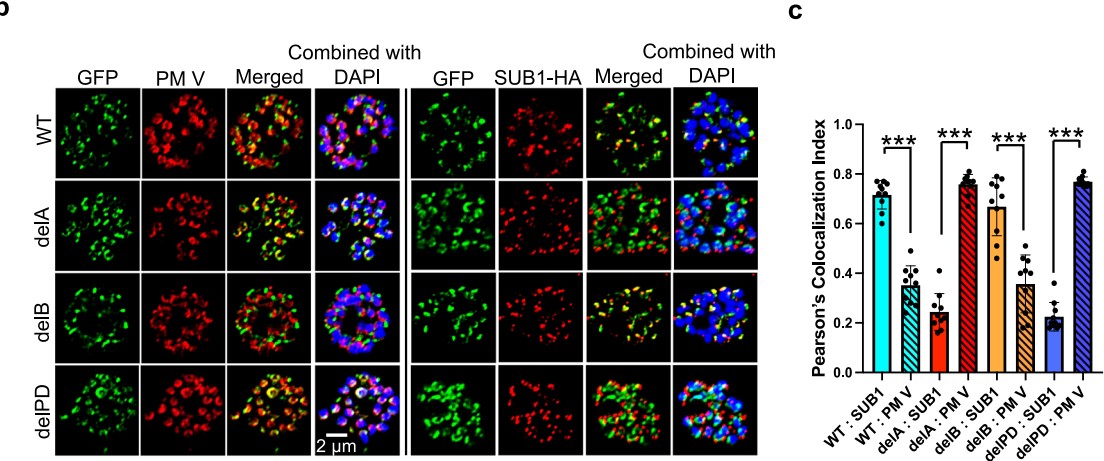

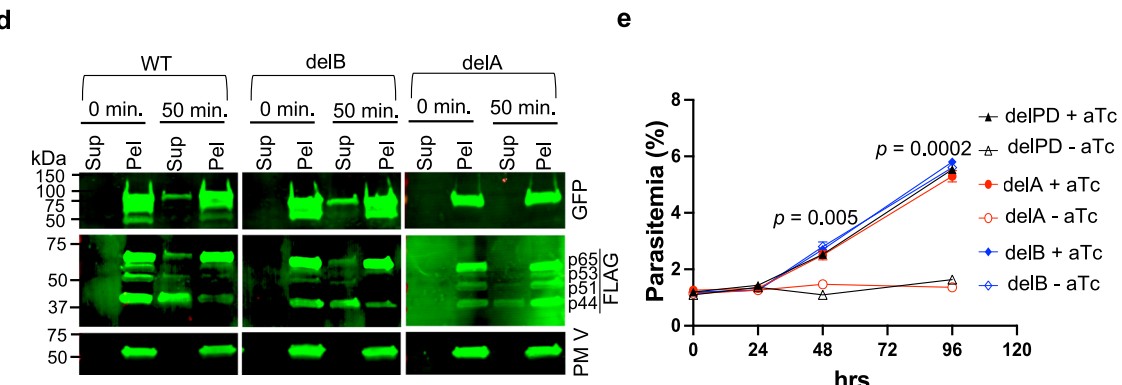

**Fig. 5 | The N-terminal half of the prodomain (PD) is critical for intracellular trafficking and functionality of PM X. a** Scheme of PD truncation constructs that are C-terminally GFP tagged and introduced into the PM X$^{apt}$ line as second copies. **b** Representative 2D confocal Airyscan images showing colocalization between the second copy PM X and either SUB1-3xHA or PM V. Scale bar: 2 μm. The 3D version of the images are shown in Supplementary Fig. 7. **c** Quantification from Supplementary Fig. 7. Experiments were repeated two times, and for each line 10 schizonts were analyzed. Mean values are shown. Error bars represent standard deviation. ***$p < 0.001$. For exact $p$ values refer to the source file. **d** Zaprinast-induced discharge of second copy WT and delB but not delA mutant PM X from

41–44 h schizonts. Samples were treated with 75 μM zaprinast for 45 min. Supernatant was then separated from the cellular pellet followed by western blots with indicated antibodies. Shown are the representative blots from two independent experiments. **e** Growth curve of indicated parasite lines showing that the delA and delPD mutants failed to rescue the growth defect due to knockdown of PM X. Mean values from three independent experiments are shown and error bars represent standard deviations. For **c** and **e** data were analyzed statistically by two-tailed Student's $t$ test. $P$ values in **e** represent comparison of delA mutant grown in presence or absence of aTc at two time points. Source data are provided as a Source Data file.

cellular compartments[8]. Similar partial colocalization was obtained using EBA-175 as a microneme marker instead of AMA1 (Fig. 8a, b). Dual labeling of PM X-3xHA-expressing terminal schizonts with either anti-RON4 (rhoptry neck marker) or with anti-RAP1 (rhoptry bulb marker) showed significantly less overlap of signals (Fig. 8c). Consistent with the Airyscan microscopy, imaging of parasites by the super-resolution structured illumination microscopy (SR-SIM) revealed regions of complete overlap between PM X and AMA1 or EBA-175 amid distinct punctate patterns (Supplementary Fig. 11).

Given the small size of the merozoites and the proximity of the organelles at the apical end of each merozoite, the regions of overlap observed between AMA1 and PM X or AMA1 and SUB1 could be a result of insufficient resolution by fluorescence microscopy. Therefore, to better understand the localization of these proteins in schizonts, we performed immunoelectron microscopy (immunoEM) on thin-section schizont samples that were pretreated with C1. This revealed vesicles that were single positive for either AMA1 or PM X-3xHA, as well as vesicular structures showing signals for both proteins (Fig. 9a). Similar observations were obtained when SUB1-3xHA schizonts were used instead of PM X-3xHA schizonts (Fig. 9b). Although we could not determine whether the single-positive vesicles bud from those that contain both proteins, our data clearly indicate common partitioning

of PM X and SUB1 with AMA1 inside the parasites, at least transiently. Despite repeated attempts by immunoEM, we did not observe PM X or SUB1 signal in the rhoptries (data not shown).

## Discussion

### Autoprocessing and activity of PM X

Aspartic proteases are canonically synthesized as proenzymes that autoactivate by self-processing at the pro-mature junction, relieving prodomain inhibition in the active site. The proenzyme form of PM X in *P. falciparum* processes itself into multiple isoforms with different N-termini and leaves some protein unprocessed. Using both recombinant enzyme and mutant parasites we mapped the PM X internal cleavage sites. Mutations at the intra-prodomain region cleavage sites could only abolish the formation of the intermediate forms, but not the fully processed form, while mutations at the pro-mature junction abolished the fully processed form but not the intermediate forms (Fig. 1e). Thus, the different truncated forms of PM X originate from independent processing events at alternative cleavage sites.

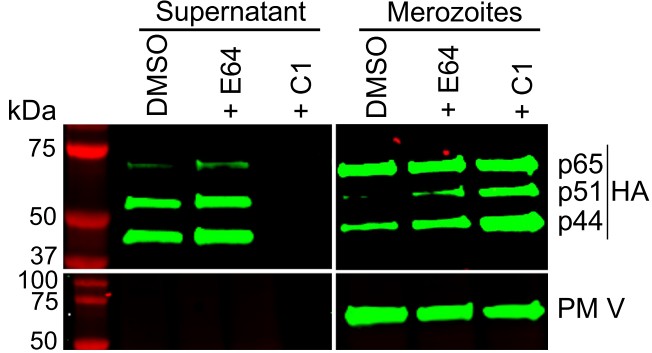

**Fig. 6 | PM X is processed in terminal secretory compartments in the parasites.** Synchronized 40–43 h schizonts expressing PM X-3xHA were treated separately with the vehicle (DMSO), E64d (10 μM) and C1 (1.5 μM) for 8 h. Samples were fractionated to separate the secreted (supernatant) components from those that remained intracellular (merozoites). PM X-3xHA was pulled down using anti-HA antibody. Western blot was done with both fractions to assess distribution of PM X. Aliquots were blotted for PM V as an intracellular control. Shown are representative blots from two independent experiments. Source data are provided as a Source Data file.

To our surprise, a quadruple mutant PM X that fails to undergo any cleavage, showed robust in vitro substrate cleavage activity, and could fully complement parasite growth in absence of endogenous PM X (Figs. 2 and 3). An ER resident protein, *Pf*ERC (*Plasmodium falciparum* endoplasmic reticulum-resident calcium-binding protein) has been implicated in the activation of PM X[24]. Knockdown of *Pf*ERC gave rise to a defect in the PM X-mediated processing step of SUB1, leading to a block in parasite egress. In those parasites, PM X accumulated in the full-length form. Our current data suggest that the observed loss of PM X functionality was likely not due to the lack of proteolytic processing.

How then is the activation of the PM X proenzyme regulated? Earlier, rPM X was shown to be active at pH 5.5[3–5]. However, it was not clear why acidic conditions are required for activity. Recently, two independent studies based on the crystal structures of the recombinant *Pf*PM X reported unique folds within its PD segment[12,25]. Importantly, a twisted loop within the N terminal part of the PD ($R^{95}$ to $G^{104}$) has been shown to occupy the substrate binding pockets in the proenzyme. Additional salt bridges between the PD and the mature domain ($R^{95}$ and $E^{433}$) and within the mature domain ($R^{244}$ with the catalytic $D^{266}$ and $D^{457}$) have been implicated in maintaining the inactive state. Like the other autocatalytic pepsin-like proteases, therefore, a drop in pH has been proposed to break the salt bridges, destabilizing the inhibitory PD, and thereby leading to PM X activation[14]. The pH dependency of rPM X for substrate cleavage supports this theory (Supplementary Fig. 6). Importantly, an N-terminal fragment of the PD remained noncovalently attached to the rest of the protease even under activation conditions (Fig. 4a, c). This indicates that the acidity-induced activation of PM X does not require physical separation of the PD.

Activity of parasite-derived PM X was also regulated by pH (Fig. 3). Nevertheless, the quadruple mutant was significantly less active than the WT PM X at pH 6.5. This suggests that while PD cleavage is not necessary, it can increase the substrate cleavage efficiency of PM X under some conditions. In parasites, however, the different cleavage mutant forms did not show a discernable defect in processing of SUB1 and AMA1 (Fig. 2b). PM X has a broad range of substrates and many of them are parasite surface adhesins[5]. The cleavage efficiency of the different mutant forms of PM X towards these adhesins was not tested here. It is possible defects in PM X-mediated processing might alter the antigenicity of different surface adhesins, and therefore might change how the surface epitopes are presented to the host immune cells. Being an intracellular pathogen that normally stays outside of its host

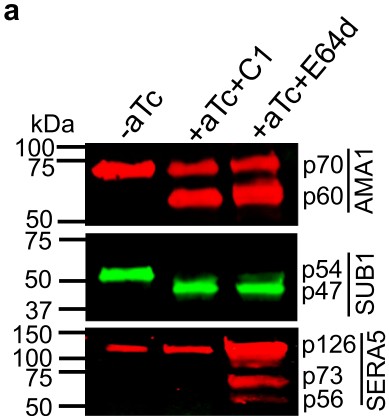

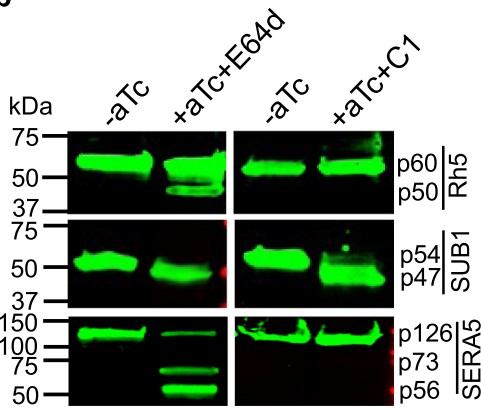

**Fig. 7 | PM X cleaves microneme substrates intracellularly, while rhoptry substrate cleavage is mediated in a post-secretion manner. a, b** PM X^apt parasites were grown in presence or absence of aTc to the schizont stage. At 44 h post invasion, either C1 (1.5 μM) or E64d (10 μM) was added to the +aTc cultures for 6 h before harvesting. Samples were blotted assess the cleavage of either AMA1 (**a** microneme substrate) or Rh5 (**b** rhoptry substrate). Processing of SUB1 and SERA5 were analyzed from the same lysates as indicators of PM X activity and exoneme discharge respectively. Each blot was repeated at least twice. Source data are provided as a Source Data file.

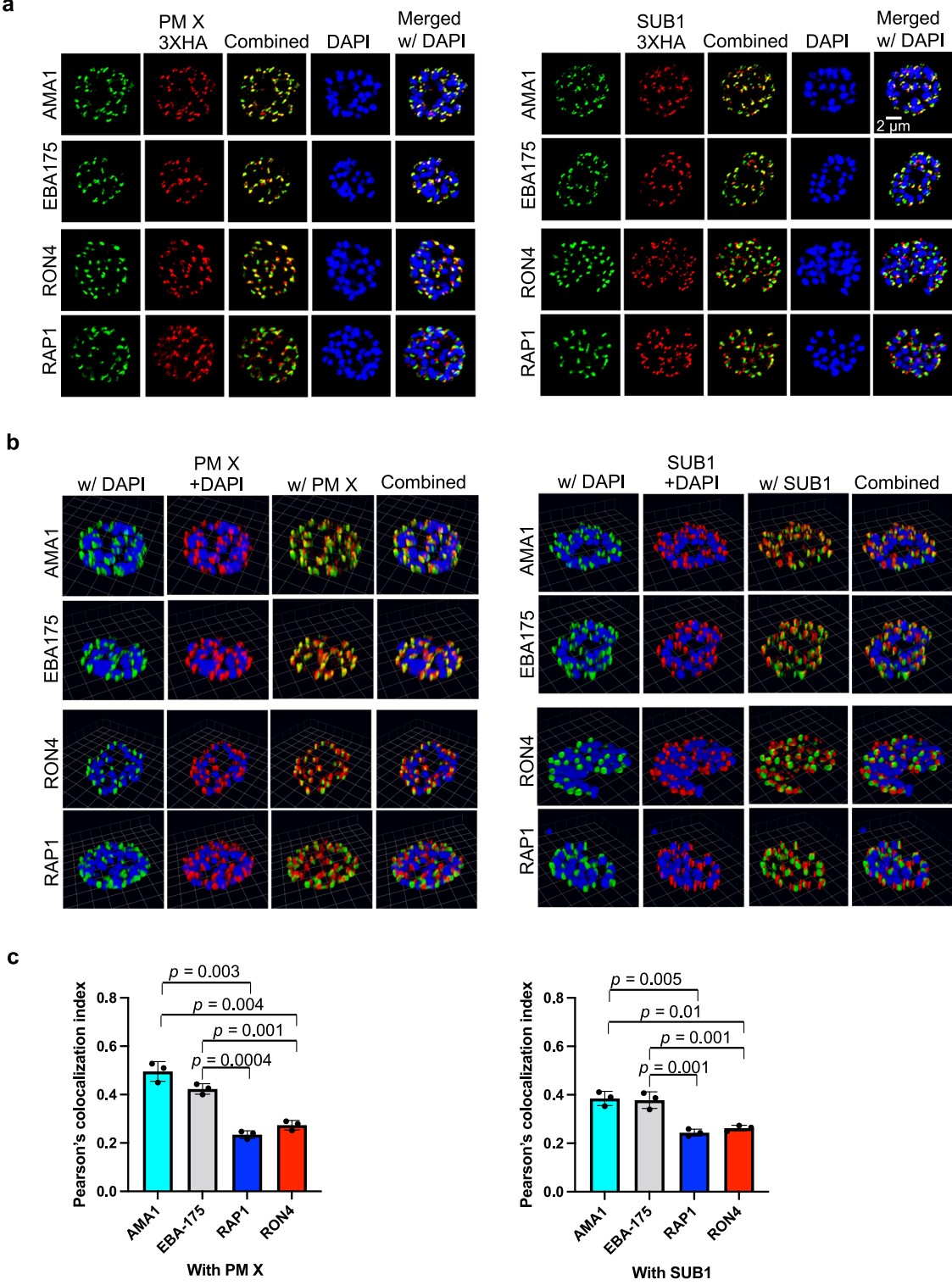

**Fig. 8 | PM X and SUB1 signals overlap significantly more with microneme markers (AMA1 and EBA175) than with rhoptry neck (RON4) or bulb (RAP1) markers in terminal schizonts.** Synchronized, C1-treated 48–50 h schizonts expressing either PM X-3xHA or SUB1-3xHA were fixed in paraformaldehyde and processed for immunofluorescence assays. After labeling with the indicated antibodies, samples were visualized by confocal Airyscan microscopy. Images in **a** are

2D snapshots. 3D reconstructions are shown in **b**. Each grid line in **b** is 1.61 μm. Shown are representative images from two independent experiments.
**c** Quantification of the 3D images from **b**. For each line, 10 schizonts were analyzed from three biological replicates. Mean values are shown and error bars represent standard deviations. Data were analyzed statistically by two-tailed Student's *t* test, *p* values are shown on the graphs. Source data are provided as a Source Data file.

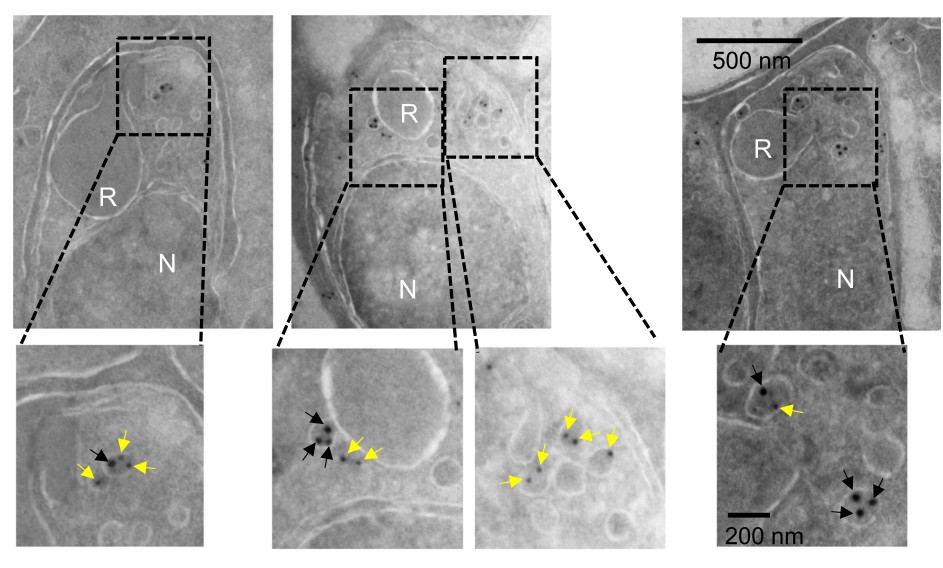

**Fig. 9 | A subset of the apically located vesicles in schizonts show colocalization between AMA1 and PM X or SUB1 by immunoEM.** Synchronized, C1-treated, 48–50 h schizonts expressing either PM X-3xHA (**a**) or SUB1-3xHA (**b**) were processed as described in the methods. Thin section samples were labeled for the indicated markers and visualized by immunoelectron microscopy. Black arrows: 18 nm beads that detect the anti HA antibody, yellow arrows: 12 nm beads that detect the anti AMA1 antibody. Sections showing apical vesicles are magnified from each image. R rhoptry, N nucleus. Shown are representative images from two independent experiments.

RBC only for short period of time between egress and re-invasion, variation in antigenicity might, therefore, cost parasite replication in vivo.

### Interaction of PM X with different substrates is spatially regulated

We showed that the PM X-mediated processing of the exoneme-localized substrate SUB1 and the microneme-localized substrate AMA1 take place exclusively inside the parasites (Fig. 7a). In contrast, the rhoptry substrate Rh5 was processed following apical organellar discharge, likely inside the PV (Fig. 7b). Given that PM X needs acidic pH for activity, this may indicate that the PV becomes acidic prior to egress. In the related apicomplexan parasite *Toxoplasma gondii*, a pH decrease in the PV towards the end of the replication cycle has been reported[26].

By using two different high-resolution immunofluorescence techniques, along with immunoelectron microscopy, we observed an intimate association between the microneme and exoneme markers (Fig. 8 and Supplementary Fig. 10). Some signal was overlapping and by EM could be seen together in vesicular structures, suggesting that the two organelles might have a common base. Based on these data, therefore, we propose that the microneme and exoneme proteins co-traffic to a precursor compartment where PM X encounters an acidic environment and subsequently cleaves its substrates. Further sorting of the proteins from such precursor compartments might result in the heterogeneity of the micronemes and exonemes.

It is still not clear why some rhoptry substrates are cleaved by PM X. In *T. gondii*, a PM IX/ X ortholog ASP3 has been shown to process both microneme and rhoptry substrates[27]. In contrast, the *Plasmodium*

genome encodes a separate rhoptry-specific aspartic protease, plasmepsin IX (PM IX)[3,4]. Importantly, PM IX and PM X have similar substrate specificity in vitro[5]. This suggests that the exposure of different substrates to these proteases in *Plasmodium* is rather complex and might be both spatially and temporally regulated. While the molecular determinants remain to be identified, sub-compartmentalization of different proteins inside the rhoptries have been documented[28,29]. It is believed that this organization regulates interaction between different proteins within the rhoptries and as well as the timing of protein secretion. Rh5 along with other Rhs are located at the very tip of the rhoptries where PM IX is excluded, and therefore, it is possible that Rh5 is not accessible to PM IX[3,4,11]. Moreover, while it remains unknown how PM X-mediated cleavage regulates Rh5 function, we have shown that PM X is secreted inside the PV (Fig. 6), where we propose that the cleavage happens. Whether PM IX is a secreted protease remains unknown.

## Methods

### Reagents
All primers were obtained from Integrated DNA Technologies. List of primers used in this study can be found in the Supplementary Table 2. Restriction enzymes were purchased from New England Biolabs. Compound 1 was purchased from MedChem Express. E64d and zaprinast were purchased from Sigma. Ni-NTA beads, anti-HA and the anti-GFP antibodies coated magnetic beads were from ThermoFisher and Chromotek respectively. The rat anti-FLAG antibody was from Novus Biologicals, the mouse-anti HA antibody was from Biolegend, the rabbit anti-HA antibody and the rabbit anti-His antibody were from Sigma, the rabbit anti-AMA1 antibody was from LSBio, the mouse anti-GFP antibody (JL-8) was from Takara. Monoclonal antibodies 2.29 (anti-RAP1) and 12.4 (anti-MSP1) were obtained from The European Malaria Reagent Repository (http://www.malariaresearch.eu). The anti-EBA175 antibody (MRA-2) was from MR-4. The mouse anti-PM V antibody was previously described[30]. Anhydrotetracycline (aTc) was purchased from Cayman Chemical. Synthetic PM X substrate peptides were from Eurogentec. List of the primary antibodies and dilutions used can be found in the Supplementary Table 3.

### Generation of plasmids
To complement the PM X knockdown, we modified the previously described pyEOE-2X attP-3X MYC vector[3]. At first, the yDHOD cassette was replaced with the human dihydrofolate reductase (hDHFR), resulting into the vector pEOE-2X attP. To make PM X expression stage-specific, we used a 1700 bp sequence located upstream of the AMA1 coding region. This promoter (pAMA1) has been successfully used for transgenic expression of other late-stage-specific genes in *P. falciparum*[10,31]. The pAMA1 was amplified from genomic DNA of the NF54 strain of *P. falciparum* using the primer pair 151 and 152 and subsequently gel extracted. The purified fragment was ligated between the AflII and XhoI restriction enzymes cut sites within pEOE-2X attP generating pEOE-pAMA1-2X attP. The ORF of PM X was amplified using the primers 153 and 154. The ORF of green fluorescence protein (GFP) was PCR-amplified from the plasmid PM2GT[32] using the primers 155 and 156. Amplified PM X and GFP ORFs were subsequently fused by Gibson assembly with the pEOE-pAMA1-2X attP that was previously digested with XhoI and EagI. This resulted in the production of the final WT complementation construct pEOE-PM X-GFP-2XattP. Subsequent site-directed mutagenesis (QuickChange lightning, Agilent) on this plasmid used primers 157, 158, 159 and 160 to generate the different PM X cleavage mutant constructs. For generating the different PD truncation mutant constructs, pEOE-PM X-GFP plasmid was subjected to site-directed mutagenesis. Mutagenesis primers 161 and 162 were used to generate the delA and delB constructs respectively. DelPD construct was made in a similar way using primer 163. Tagging the endogenous ORF of SUB1 with 3xHA was carried out using previously published plasmids[3]. The different cleavage mutant constructs for recombinant PM X were made from the parental construct using the primers shown in Supplementary Table 2.

### Parasite culture and transfection and synchronization
NF54attB parasites[33] and resultant transgenic strains were cultured in human red blood cells as previously described[34]. PM X[apt] parasites[3] were maintained in presence of 100 nM aTc and 2.5 µg/ml BSD. For complementation of PM X, plasmids carrying WT or mutant PMX coding sequences were independently co-transfected with a Bxb1 integrase plasmid[35] into the PMX[apt] parasites. Parasites were selected with 5 nM WR, together with 2.5 µg/ml BSD and 1 µM aTc. The integration of these plasmids into the PMX[apt] strain was confirmed by PCR using the primer pair 164 and 165. For tagging the 3' end of SUB1, the NF54attB parasites were cotransfected with the yPM2GT-SUB1-3xHA and the pAIO3 guide plasmids. The transfectants were selected in the presence of 12.5 µg/ ml DSM1. Integration of the plasmid was verified by PCR.

Highly synchronous ring-stage parasites were obtained as follows. High-parasitemia schizont cultures were passed through MACS LD magnet columns (Miltenyi Biotec) and schizonts were collected. These were then added to fresh uninfected RBCs resuspended in warm culture media. The cultures were shaken at 80 RPM for 3 h and the resulting parasites were synchronized using 5% sorbitol as described before[36].

### Parasite growth assay
For growth curves, synchronized ring-stage parasites were washed extensively to remove residual aTc and were diluted to an initial 1% parasitemia at 2% hematocrit. Cultures were split and aTc was added to appropriate wells. Parasitemia was determined every 24 h using flow cytometry as described previously[3].

### Natural and zaprinast-induced parasite egress assays
To follow natural parasite egress, 41–44 h schizonts expressing PM X-3xHA were MACS- purified and treated with either vehicle (DMSO) or with E64d (10 µM) for 8 h. A third sample was treated similarly with the PKG inhibitor C1 (1.5 µM) to block discharge from the apical organelles[19]. Following incubation, the discharged proteins (in the culture supernatant) were separated from those that remained intracellular by centrifuging the cultures at 10,000 × *g* for 10 min. PM X was subsequently pulled down using anti-HA beads and subjected to western blot.

Zaprinast-induced organellar discharge was performed as described previously[19]. Briefly, 41–44 h synchronized parasites were MACS purified and transferred to albumax-free medium. Zaprinast (75 µM) was added to the cultures and samples were either harvested immediately (0 min) or after 45 min of incubation. Culture supernatants were separated from the intracellular fractions as mentioned. Endogenous or second copy PM X was pulled down from both fractions using anti-FLAG and anti-GFP antibodies respectively, followed by western blot.

### Brefeldin A sensitivity assay
To determine the effect of brefeldin A (BFA) on the processing of PM X, a pulse-chase assay was performed as before[37]. Briefly, synchronized cultures of schizont-infected RBCs were washed once in RPMI without methionine and cysteine, resuspended in labeling medium (RPMI without methionine and cysteine) containing 250 µCi/ml [35S] methionine-cysteine (PerkinElmer) and incubated at 37 °C for 5 min. Cultures were either harvested immediately or maintained in label-free complete medium containing 10 µg/ml cycloheximide for 60 min at 37 °C before harvest. During incubation, BFA (5 µg/ml) was added to one culture. Subsequently, PM X-3xHA was pulled down, resolved by SDS-PAGE and subjected to autoradiography.

## Expression and purification of recombinant PM X (rPM X) from mammalian cells

The rPM X expression vector for mammalian cells was a gift from UCB Pharma, USA. In this construct, a mammalian recodonized version of PM X was C-terminally tagged with 8x-histidine (8x-His) and the endogenous signal peptide at the N-terminal end was replaced with a mammalian signal peptide. For protein expression, the Freestyle 293-F cells (Thermo Fisher) were transfected with a mixture of DNA and polyethylenimine (1:3). The rPM X was expressed as a secreted protein. 72 h post transfection the supernatant was harvested. The diluted supernatant (1: 3 in 50 mM Tris-Cl pH 8.0, 100 mM NaCl, 10 mM imidazole) was purified using Ni-NTA resin (ThermoFisher). After extensive wash in the same buffer, the protein was eluted with 50 mM Tris-Cl pH 8.0, 100 mM NaCl, 400 mM imidazole. The eluted sample was then buffer exchanged using an Amicon ultra 3 kDa molecular wt. cutoff filter (Millipore, USA) and finally resuspended in 50 mM Tris pH 7.5, 100 mM NaCl. If not used immediately, 1% glycerol was added and the preparation was stored in −80 °C.

## Circular dichroism (CD) spectroscopy assay

To examine the secondary structure of rPM X, samples were diluted to 35 ug/ ml in 10 mM potassium phosphate buffer with indicated pHs. CD spectra were recorded on a JASCO-J715 polarimeter (JASCO, Tokyo, Japan) over the wavelength range 185–325 nm in a 1-mm path length quartz cuvette using a step size of 0.5 nm, a slit bandwidth of 1.0 nm, and a signal averaging time of 1.0 s. For each wavelength three scans were performed. AVIV software was used for background subtraction. Molar ellipticity $[\theta]$ was calculated using the formula: $[\theta] = \theta(mdeg)/10 \cdot C \ (mol/lit) \cdot l(cm)$, where C is the concentration of the sample, l is the path length. For estimation of secondary structure composition, the online server K2D3 was used.

## In vitro substrate cleavage assay

The ability of rPM X to cleave fluorogenic substrate peptides was performed as described[4,5]. Briefly, 25 ng of rPM X was incubated with 1 μM DABCYL-HSFIQEGKE-E-EDANS (Rh2N peptide) in 200 μl of PM X activity buffer (25 mM MES, 25 mM Tris, pH 5.5) for 40 min at 37 °C. As a negative control, a point mutant form of the peptide (DABCYL-HSFAAEGKE-E-EDANS) that is not cleaved by PM X was used as a substrate. To inhibit PM X activity, 1 μM CWHM-117[3,38] was added to samples at pH 5.5. Each sample was analyzed in triplicate using a Synergy HTX microplate reader at 340 nm wavelength. Readings were taken at 1 min intervals. PM X activity was measured as an increase in the relative fluorescence units (RFUs) over time. Shown are the final RFU values after 40 min of reaction.

To determine the activity of parasite-derived PM X, synchronized 42–45 h schizonts ($2 \times 10^7$ cells per sample) expressing PM X-GFP were lysed in ice-cold native lysis buffer (50 mM Tris, pH 7.5, 150 mM NaCl, 1% NP-40). PM X-GFP was pulled down from whole cell lysates using anti-GFP antibody-coated magnetic beads (Chromotek) according to the manufacturer's recommendations. Following washes (4×) in the lysis buffer, beads were incubated with the Rh2N substrate peptide (1 μM) in 200 μL of PM X activity buffer. Fluorescence readings were taken either immediately or following 1 h incubation in 37 °C.

## Liquid chromatography/Mass spectrometry (LC/MS)

To determine the cleavage sites of the processed products in rPM X, the indicated bands in Fig. 1A were excised from the gel and were subjected to LC/MS analysis. Briefly, the gel samples were destained, reduced and alkylated prior to trypsin digestion. Digest was aspirated and combined with further extraction of the gel piece with 60% acetonitrile (ACN) in 1% trifluoroacetic acid (TFA). The extracted peptides were dried down and each sample was resuspended in 10 μL 5% ACN/ 0.1% formic acid (FA). 5 μL was analyzed by LC-MS with a Dionex RSLCnano HPLC coupled to an Orbitrap Fusion Lumos (Thermo

Scientific) mass spectrometer using a 2 h gradient. Peptides were resolved in a 75 μm × 50 cm PepMap C18 column (Thermo Scientific) with following gradient: Time = 0–4 min, 2% B isocratic; 4–8 min, 2–10% B; 8–83 min, 10–25% B; 83–97 min, 25–50% B; 97–105 min, 50–98%. Mobile phase consisted of A, 0.1% FA; mobile phase B, 0.1% FA in ACN. The instrument was operated in the data-dependent acquisition mode in which each MS1 scan was followed by Higher-energy collisional dissociation (HCD) of as many precursor ions in 2 s cycle (Top Speed method). The mass range for the MS1 done using the FTMS was 365–1800 m/z with resolving power set to 60,000 at 400 m/z and the automatic gain control (AGC) target set to 1,000,000 ions with a maximum fill time of 100 ms. The selected precursors were fragmented in the ion trap using an isolation window of 1.5 m/z, an AGC target value of 10,000 ions, a maximum fill time of 100 ms, a normalized collision energy of 35 and activation time of 30 ms. Dynamic exclusion was performed with a repeat count of 1, exclusion duration of 30 s, and a minimum MS ion count for triggering MS/MS set to 5000 counts.

## Data analysis

All MS/MS samples were analyzed using Mascot (Matrix Science, London, UK; version 2.5.1.0). Mascot was set up to search against provided sequence along with common contaminants. The digestion enzyme was set as semiTrypsin (to account for nontryptic termini). Mascot was searched with a fragment ion mass tolerance of 0.60 Da and a parent ion tolerance of 10 ppm. Oxidation of methionine, carbamidomethylation of cysteine, and acetylation of N-terminal of protein were specified in Mascot as variable modifications. Scaffold (version Scaffold_4.8.2 Proteome Software Inc., Portland, OR) was used to validate MS/MS based peptide and protein identifications. Peptide identifications were accepted if they could be established under 1% FDR by the Peptide Prophet algorithm[39] with Scaffold delta-mass correction. Protein identifications were accepted if they could be established at greater than 99.0% probability and contained at least 2 identified peptides. Protein probabilities were assigned by the Protein Prophet algorithm[40]. The mass spectrometry proteomics data have been deposited to the ProteomeXchange Consortium via the PRIDE[41] partner repository with the dataset identifier PXD035172 and 10.6019/PXD035172.

## Size exclusion chromatography

For gel filtration, rPM X samples were pretreated in acidic PM X activation buffer (25 mM MES, 25 mM Tris, pH 5.5) or in the same buffer with a pH 7.5 for 30 min at room temperature. Before sample runs, the Superdex 200 10/300 column (GE Healthcare) was equilibrated in the same buffer. Following runs, the samples were collected as 0.5 ml fractions. Fractions of interest (as indicated in Fig. 4b and Supplementary Fig. 4a) were further concentrated using an Amicon ultra 3 kDa mol. wt. cutoff filter. Finally, samples were resolved by SDS-PAGE and the gel was stained with Coomassie dye. For gel filtration under denaturation, every step was done as mentioned except that the buffers were supplemented with 6 M guanidine hydrochloride. For denaturation, sample was treated with 6 M guanidine hydrochloride and heated at 60 °C for 45 min before loading onto the size exclusion column. As a control for acidity-induced dissociation of subunits, human hemoglobin A tetramers were treated in the same buffers as rPM X followed by gel filtration.

## Dynamic Light Scattering (DLS)

To determine the size distribution profile of rPM X in pH 5.5 buffer, a DLS experiment was performed with the indicated fractions obtained from size exclusion chromatography. Sample concentration was adjusted to 0.5 mg/ ml. Briefly, 40 ul of sample was taken in disposable polystyrene cuvettes and subsequently the light scattering was measured in a Malvern ZEN3600 instrument fitted with a 632 nm laser

illuminating source. Each sample was measured in duplicates. Following sample run, all analysis was performed using the Zetasizer 7.13 software. PyMOL 2.4.1 was used to determine the diameter of the recombinant PM X (PDB: 7RY7).

### Western blots

For PM X knockdown western blots (Fig. 2b), following synchronization, each culture was split into two plates. To one plate, aTc was added, and to the other equal volume of vehicle (DMSO) was added. 46 h post invasion, schizonts were collected by saponin lysis and stored at −80 °C until further processing. Samples were subsequently lysed in RIPA buffer supplemented with protease inhibitor cocktail and hemozoin was removed by centrifugation. Cell lysates were mixed with 4x sample loading buffer and heated at 99 °C for 5 min. Fractions of equal cell equivalents were subjected to SDS-PAGE and immunoblotting.

For processing inhibition assays (Fig. 7), PM X[apt] parasites were synchronized and grown up to 44 h of age, as above. The infected RBCs from +aTc were then separated by MACS and resuspended in a small volume supplemented with either E64d (10 μM) or with compound 1 (1.5 μM) for the next 6 h. Cultures were then separated from the culture supernatant and the pellet fractions were boiled with 2× sample loading buffer.

Primary antibodies included rabbit anti-AMA1 (1:500), rabbit anti-SUB1 (1:1000), rabbit anti-HA (1:1000), rabbit anti-Sera5 (1:1000), mouse anti PM V (1:500), rabbit anti-Rh5 (1:500), mouse anti-GFP (1:1000), mouse anti-his (1:1000), and rat anti-FLAG (1:1000). For all, appropriate IRDye conjugated secondary antibodies were used at 1:10000 dilution. Blots were visualized on an Odyssey imaging system (Licor).

### Immunofluorescence assays

For IFAs, cells were fixed as described previously[42]. Briefly, synchronized and C1-treated mature schizonts were fixed in 3.7% paraformaldehyde for 15 min and blocked in 3% BSA in PBS overnight at 4 °C before antibody staining. The antibodies used for IFA were: rabbit anti-HA (1:500), mouse anti-HA (1:500), mouse anti-GFP (1:500), mouse anti-MSP1 (1:500), rabbit anti-AMA1 (1:500), mouse anti-RAP1 (1:500), rabbit anti-EBA175 (1:500), rabbit anti-RON4 (1:500) and mouse anti-PM V (1:50). The secondary antibodies were used as 1:2000 dilutions and were conjugated to Alexa Fluor 488 or 546 (Life Technologies). Cells were mounted with ProLong and 4′,6′-diamidino-2-phenylindole (DAPI) (Invitrogen) and imaged using a Zeiss Imager M2 Plus wide field fluorescence microscope, using a 63x objective and the Axiovision 4.8 software for epifluorescence imaging.

For confocal imaging, samples were analyzed with a Zeiss LSM880 laser scanning confocal microscope with Airyscan (Carl Zeiss Inc. Thornwood, NY). A Plan-Apochromat 63X (NA 1.4) DIC objective and ZEN black software (version 2.1 SP3) were used for image acquisition. The image analysis software Volocity (version 6.3) (PerkinElmer, Waltham, MA) was used for 3-dimensional rendering of Z slices acquired through the depth of the parasites. For SR-SIM microscopy, images were captured with a Nikon SIM super resolution microscope under a 100x objective using the Nikon NIS software. Image processing and display were performed using Zen Blue. Adjustments to brightness and contrast were made for display purposes.

### Immunoelectron microscopy

Samples for immunoEM were prepared as before[3]. Briefly, the infected RBCs were fixed in 4% paraformaldehyde (Polysciences Inc., Warrington, PA) with 100 mM PIPES/0.5 mM $MgCl_2$, pH 7.2 for 1 h at 4 °C. Samples were then embedded in 10% gelatin and infiltrated overnight with 2.3 M sucrose/20% polyvinyl pyrrolidone in PIPES/MgCl2 at 4 °C. Samples were trimmed, frozen in liquid nitrogen, and sectioned with a

Leica Ultracut UCT cryo ultramicrotome (Leica Microsystems Inc., Bannockburn, IL). 50 nm sections were blocked with 5% FBS/5% NGS for 30 min and subsequently incubated with primary antibodies for 1 h, followed by secondary antibodies conjugated to 12 nm or 18 nm colloidal gold (1:30) for 1 h. Sections were washed in PIPES buffer followed by a water rinse, and stained with 0.3% uranyl acetate/2% methyl cellulose and viewed on a JEOL 1200EX transmission electron microscope (JEOL USA, Peabody, MA) equipped with an AMT 8 megapixel digital camera (Advanced Microscopy Techniques, Woburn, MA). All labeling experiments were conducted in parallel with controls omitting the primary antibody, which were consistently negative at the concentration of colloidal gold-conjugated secondary antibodies used in these studies.

### Statistical analysis

Unless specified otherwise, assay values in all figures are averaged from three independent repeats and error bars represent standard deviations. Differences were assessed by the two-tailed Student's $t$ test using the Microsoft Excel software. For quantification of colocalization in Figs. 5c and 8c confocal Airyscan images were analyzed by ImageJ and a Pearson's colocalization coefficient was determined for the indicated antigen pairs. $P$ values indicating statistical significance were grouped in all figures. All graphs were prepared using GraphPad Prism 8.

### Reporting summary

Further information on research design is available in the Nature Research Reporting Summary linked to this article.

### Data availability

Authors can confirm that all relevant data are included in the paper and/ or its supplementary information files. Source data are provided as a Source Data file. Additional supplementary information for LC/ MS are included as Supplementary Data 1, Supplementary Data 2 and Supplementary Data 3. The lists of primers and the details of the primary antibodies used are included as Supplementary Table 2 and Supplementary Table 3 respectively. The crystal structure of PM X that was used in this study is deposited in the protein data base (PDB) under PDB ID: 7RY7. The mass spectrometry proteomics data have been deposited to the ProteomeXchange Consortium via the PRIDE partner repository with the dataset identifier PXD035172 and 10.6019/ PXD035172. Source data are provided with this paper.

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

## Acknowledgements
This work was supported by grant AI138447 from the National Institute of Allergy and Infectious Diseases (NIAID) to D.E.G. We thank Dr. Wandy Beatty (WUSTL) for immune-electron microscopy, the Molecular Microbiology Imaging Facility at WUSTL for confocal Airyscan microscopy and the Washington University Center for Cellular Imaging (WUCCI) for SR-SIM microscopy, the Washington University nano research facility for Dynamic Light Scattering experiment, the Proteomics & Mass Spectrometry Facility at the Danforth Plant Science Center for LC/MS data acquisition and analysis, Dr. Bob Krantz (WUSTL) and Dr. David Sibley (WUSTL) for usage of the JASCO-J715 polarimeter and the Synergy HTX microplate reader respectively, Dr. Michael J. Blackman (The Francis Crick Institute, London) for anti-SUB1 and SERA5 antibodies, Dr. Jean-François Dubremetz (University of Montpellier, France) for anti-RON4 antibody and Dr. Eizo Takashima (Ehime University, Japan) for anti-Rh5 antibody. For rPM X production, the codon-engineered PM X gene and protocols were provided by Don Lorimer (UCB Pharma). We also thank Barb Vaupel for assistance with cloning, Dr. Darya

Urusova (WUSTL) for gel filtration assays, Dr. Eva Istvan, Dr. Sebastian Nasamu and Dr. Alexander Polino for useful suggestions.

## Author contributions

S.M. and D.E.G. conceived and designed the study. S.M. performed research, acquired, and analyzed data. S.N. and E.S. acquired data. S.M. and D.E.G. wrote the manuscript.

## Competing interests

All authors declare no competing interests.
