## [Peer Review File · Nature Communications]

Reviewer comments, first round

Reviewer #1 (Remarks to the Author):

Malaria parasites invade and escape (egress) from red blood cells, destroying the host cells in the process. In this study, the authors examine the maturation, trafficking and function of a parasite aspartic protease called plasmepsin 10 (PM X) that has recently been shown to play key roles in the maturation of several parasite proteins involved in egress and invasion. Substrates of PM X include several parasite proteins that are stored prior to egress in different types of subcellular secretory organelles of the parasite, known as exonemes, micronemes and rhoptries; these organelles are thought to undergo discharge at different points in the egress/invasion pathway. Although it has previously been shown that maturation of PM X involves autocatalytic proteolytic cleavage (similar to many other proteases), whether or how this leads to PM X activation is unknown, and how the enzyme accesses its various, differentially-localised endogenous substrates is also unclear. So this work addresses fundamental questions relevant to the biology of an important pathogen.

The authors approach these questions in two stages. First they map the proteolytic cleavage sites in PM X, exploiting a recombinant form of PM X which the authors believe to be processed in manner similar to that of the parasite protein. They validate these data by introducing mutations into the parasite PM X gene designed to block the internal cleavage events, and show that these mutations are able to prevent processing in the parasite. Unexpectedly (by analogy with many other proteases), they find that all the mutants forms of parasite PM X – including a multiple mutant that appears unable to undergo any processing – are trafficked normally and are functional both in terms of supporting parasite growth and (for the multiple mutant) in protease assays. They further present evidence that the N-terminal PM X prodomain segment is important for trafficking of the enzyme. Next, they address substrate access. Using a kinase inhibitor to arrest the discharge of secretory organelles, they show that PM X cleaves microneme and exoneme substrates before their discharge. In contrast, cleavage of known rhoptry substrates is concluded to occur following rhoptry discharge. Based on immunofluorescence imaging and immunoEM analysis, they suggest that PM X is at least transiently co-localised in vesicular structures with proteins that are thought to eventually accumulate in distinct organelles.

Some of the work is well done and overall the study may reveal some valuable new insights into an intriguing aspect of parasite cell biology. However, several aspects of the experimental work lack rigour and/or are particularly poorly and often inaccurately explained, leading to real confusion on the part of this reviewer in several instances. There is a general lack of thoroughness and attention to detail and some of the data are of inadequate quality. The figures and figure legends are often superficial, often incomplete and unclear. As a result, several conclusions are insufficiently well supported by the data, leading to potentially misleading statements.

PM X processing

1. Line 68, Introduction. Here we are told that PM X 'has a long prodomain (PD) and a C-terminal catalytic domain'. This statement is not referenced, and the rationale for referring to the N-terminal segment as a prodomain is not provided. Given the focus of this manuscript, much better clarification is required here.
2. Lines 95-96 and Fig 1. The 16 kDa recombinant PM X species identified on Coomassie-stained SDS PAGE gels of the purified protein is claimed to represent 'the N-terminal prodomain fragment of rPM X', yet the red arrowed line in Fig 1b only spans the N-terminal one third or so of the prodomain as marked. So presumably the authors are suggesting that the 16 kDa species is not the entire prodomain but a fragment of it. However, this is not actually stated, and the basis of the prodomain/mature domain layout as drawn in Fig 1b is simply not adequately explained. The term 'semi-pro' is then introduced at the end of this paragraph with no explanation of what this actually means. This is careless and confusing, and the authors' conclusions as to how the protein is processed and the basis of the schematic shown are never properly explained. Please re-write this entire section to more accurately state the findings. How do the authors know that the 44 kDa species (which is not marked as such on the gels) is the 'mature' isoform of PM X? Please use such terms appropriately and accurately and in a manner that is supported by the data.
3. Is it possible that the recombinant proteins are modified by glycosylation, which may prevent

more conversion to lower forms? Notably, the p51 rPM X intermediate reported previously by other groups as a rPM X intermediate is completely absent from this protein preparation, the full length being detected only when a PMX inhibitor was present (see PMID: 32109369). Why do the authors not comment on this? Use of the available X-ray structure of PMX (PDB id 6ORS) could be used in Fig 1 to better explain the different forms of the enzyme.

4. The Fig 1b legend refers to the marked putative cleavage sites as 'autocatalytic tetrapeptides'. This is an inaccurate way of referring to these sequences, which should instead be referred to as the sequences flanking the putative cleavage sites or scissile bonds (the double slashes at the scissile bond might better be replaced by a downward-pointing arrow or hyphen, and Schechter and Berger nomenclature should be used throughout as conventional). Similarly, the text referring to the sequences for which coverage was obtained by mass spectrometry is poorly written. Please re-write. The red and blue lines with terminal arrow heads refer to the bands in Fig 1a marked with red and blue arrows, but what does the black line in Fig 1b refer to, and how was the SD-IQ cleavage site identified?

5. Lines 103-112. Here the various mutations are described and the results presented, but several points are poorly explained. The previously-described cleavage specificity of PM X should be described in the Introduction, rather than just referred to obliquely here (lines 107-108). Line 107 also refers to a tetrapeptide motif that appears to be marked by a black line on Fig 1b (but is not mentioned in the figure legend – see point above). Is there any experimental evidence that cleavage actually occurs at this site? This was not clear to this reviewer. The authors must find a much clearer, less ambiguous way of describing their results here. Overall, the results presented in Fig 1 would greatly benefit from an additional schematic (similar to those in Fig 5a) showing the positions in the PM X sequence of the various mutations and how they altered cleavage. As it stands, it is difficult to assess the effects of the various mutations on PM X processing.

6. Regarding the main mutagenesis work, the parasite-expressed tagged PM X second copies would better have been designed to possess a small tag (HA or 3xHA) to prevent artefacts due to the nature of the tag. In the manuscript the authors mentioned the use of a HA-tagged PM X parasite second copy but this construct was not used for the mutagenesis work, a GFP tagged version being used instead. GFP is known to dimerise in solution and can lead to artefactual results. The work presented here should be repeated with the PMX HA tagged version to confirm or not the results obtained with the GFP tagged version and confirm whether or not full length unprocessed PMX is active on native substrates.

7. Recombinant and native (parasite) forms of PMX should ideally be compared side by side on SDS-PAGE in order to establish whether the fate of the recombinant protein genuinely mimics that of the parasite proteins. Although the use of protein tags will undoubtedly impact on protein migration within the gel, use of small tags would still allow comparison of the processing profiles. Fig 3 would benefit from an SDS-PAGE showing the PMX pull downs corresponding to the wild type, Quad and D266G mutant.

Role of the prodomain in PM X activity and activation

1. In section 2.3, the authors use CD to detect structural differences between purified PM X at pH 7.5 and pH 5.5 and refer to 'acidity-induced activation' of PM X. No secondary structural data are provided to support the suggestion that the CD spectra changes were acidity-induced. CD results should be presented as % composition of secondary structure (alpha, beta, turn). This information is missing in the paper. The authors do not discriminate between activity at low pH and activation by low pH, nor do they prove any causal link between the structural changes they observe and enzyme activity. The authors do not appear to consider the possibility that the complex between the 16 kDa form (which is only a fragment of the prodomain anyway, based on Fig 1 data and the schematic shown in Fig 1b) and the rest of the protein may be non-stoichiometric. If it is, it remains possible that the only a fraction of the purified recombinant PM X that lacks the 16 kDa fragment is enzymatically active. Taken together, their conclusion that the 'in vitro activity of rPM X is regulated by acid-induced structural changes' (the subtitle used for Results section 2.3) is a potentially misleading over-interpretation of their data and is not warranted.

2. Line 155. after acid-induced 'activation' of recombinant PM X, there is still a lot of residual full-length unprocessed form. Why is that? Could glycosylation be present? A recombinant version of the quadruple mutant could also be expressed HEK cells to test its capacity to cleave the Rh2N fluorescent peptide substrate.

3. Fig4a-c. Dynamic Light Scattering (DLS) or mass photometry techniques should be used to determine the various states of recombinant PM X in solution, and the results added into this

figure. This would clarify the presence of a complex between recombinant PMX 44 kDa and the 16 kDa species and indicate whether there is any uncomplexed recombinant PMX 44 kDa form in solution. Peak 1 should be split into several elution fractions as done in the denaturing conditions and the fractions submitted to DLS and SDS-PAGE. Unfortunately, a native gel cannot give a definitive answer as to whether any of the recombinant PMX 44 kDa is in an uncomplexed form.

Trafficking and function of PM X

1. The immunofluorescence data provided in Fig 5b are unfortunately not compelling in their current form, as the resolution of these images is not sufficiently high to enable discrimination between the various subcellular organelles and the parasite ER (as acknowledged by the authors themselves on lines 241-243). These experiments should be repeated using super-resolution (as in Fig 8) to better define the localisation of the various prodomain mutants. The assertion that the prodomain is required for PM X folding (lines 181-182) is completely speculative and is not addressed by any of these experiments.

2. The experiment described in Fig 6 simply shows that PM X is not released from exonemes by treatment with the C1. Others have previously shown that exoneme discharge is blocked by this PKG inhibitor, so this is not an unexpected finding. Fig 6 is also the first image in the manuscript to label the various PM X species (P67, P54 and P45). For consistency, similar labelling should be applied throughout where warranted. The pulse-chase experiment shown in Supplementary Fig 4 does indicate that PM X undergoes some of its processing in a post-ER compartment. However, it shows no more than that. It would have provided much more information to carry out a much longer chase (ideally as a time-course) in order to better visualise conversion of the PM X precursor and obtain data on the kinetics of this. What is the basis of the statement that BFA induces accumulation of newly synthesized parasite proteins in a 'fused ER/Golgi' compartment? This statement is not referenced.

3. The question over how PM X targets endogenous protein substrates that are themselves localised in different organelles is an interesting one. The experiments presented in Fig 7 were designed to examine this, but are not wholly convincing as they stand, as in Fig 7a SUB1 appears not to undergo processing whether the parasites are aTc-treated or not. Can the authors explain this? The partial co-localisation between the various proteins analysed in Fig 8 is interesting, but does not suggest that PM X comes together with these proteins in a 'precursor compartment'. Whilst it is fine to speculate on this possibility in the Discussion (lines 308-315), no definitive data are provided for this so the suggestion should be removed from the Abstract. The experiments described in Fig 8 and Supplementary Fig 6 should be repeated using schizonts in which egress has been blocked for several hours with C1, to assess whether localisation differences are due to differences in maturity of the schizonts.

Other issues:

Fig 1b. The AMA1 processing forms appear to be labelled 'P7' and 'P6' rather than 'P70' and 'P60'. Please correct.

Please amend and improve the schematics shown in Fig 5a. Why is delB not aligned with the other mutants? The current layout is sub-optimal and confusing.

Fig 7. The SUB1 conversion pattern is not clear. Arrows should be used to indicate the p54 and P47 forms.

Fig 9. Arrows should be used to indicate the type of beads visible on the micrographs. It is presently unclear what the reader is looking at.

Supplementary Fig 3. The legend here should be improved. Please remind the reader what the plotted samples are.

Supplementary Fig 6S. IFAs are of poor quality. Inset "regions of interest" are not convincing.

Reviewer #2 (Remarks to the Author):

Review of "Maturation and substrate processing topography of the Plasmodium falciparum invasion/egress protease plasmepsin X." by Sumit Mukherjee, Eashan Sharma and Daniel E. Goldberg

SUMMARY

The author aimed to address how PM X processing regulate its trafficking, activity, and substrate specificity in the malaria parasites. They demonstrated that different PM X fragments arise from processing events that are independent of each other, using parasites with mutations in both the cleavage sites of either the intra-prodomain region or pro-mature junction, or all four sites (Fig. 1). They also showed that PM X proenzyme is activated in acidic conditions (pH 5.5), likely due to disruption of salt bridges that leads to the subsequent destabilisation of the prodomain (PD), and this is supported by the pH dependency of recombinant PM X (rPM X) for cleavage activity (Supplementary Fig. 2). The author indicated that PM X proenzyme activation does not require physical separation of the PD (Fig. 4a-c) and PD cleavage increase substrate cleavage efficiency under some conditions, specifically at pH 6.5 (Fig. 3). This also illustrates that PM X processing is dispensable for its function in parasites (Fig. 3). Defects in SUB1 and AMA1 processing in mutant parasites is, however, not discernible to that of wild type (Fig. 2b). The author also showed that PM X substrate processing of SUB1 (exoneme-localised) and AMA1 (microneme-localised) occurs only within the parasite, while Rh5 (rhoptry-localised) occurs within the parasitophorous vacuole (PV) (Fig. 7). They also noted that as acidic conditions are required for PM X activity, it is likely that PV becomes acidic before egress. The author proposed that PM X processing of microneme and exoneme proteins occur together in a precursor compartment and subsequent sorting of proteins from such compartments lead to microneme and exoneme heterogeneity, due to the co-localisation observed using epifluorescence microscopy (Fig. 8 and Sup. Fig. 6.). Lastly the author suggested that PM IX and X processing is spatially and temporally regulated, citing evidence of sub-compartmentalisation of different rhoptry proteins. Specifically, Rh5 and other Rh5 located at the rhoptry tip is inaccessible to PM IX as it is excluded. The author propose PM X cleavage of Rh5 occurs within the PV, where the protease is shown to be secreted (Fig. 6). Overall, the introduction was very clear and provided all the necessary background. The figures are well designed, the data is strong, and the result section results in a complex but easy-to-follow story. The methods are very thorough. The discussion is interesting and completes the results nicely.

MAJOR COMMENTS

Fig 1. There is a lot of information in this fig but it takes several readings to work out what is going on and perhaps the figure needs clearer labelling. For example;

In 1a, where do the bands not labelled with arrows fit onto the protein map in 1b?

The labels on the 1b protein map do not correspond well with the labels on parts c, d and e.

With respect to the protein map of 1b (or perhaps show a new map of the GFP fusion protein) where does p90, p77 and p68 fit in 1e?

Line 239. Author stated that there is "no overlap of signals" between PM X-3xHA and RON4 / RAP1 signal, but figure shows some degree of overlap. It is a bit odd that Pearson's coefficients were used to support the conclusions of Fig 5 and not Fig 8. The title of Fig 8 should be changed too because it appears the PMX does partially colocalise with the rhoptry markers but not as much as the microneme markers.

Fig 9. Can the EM panels be labelled with the antibodies used like the fluorescence images? Are we meant to be able to tell the difference between the colloidal gold particles? Perhaps consider indicating the particle size with arrows.

Note, I attempted to view the excel files containing the mass spec data but my computer advised me not to open them because they were corrupted.

MINOR COMMENTS

Line 156. The word "blood stream" is usually written as one word "bloodstream".

Line 265. The word "prodomain" is incorrectly spelled as "prodomian".

Line 287. The author may be referring to Supplementary Fig. 2, instead of Supplementary Fig. 3

Lines 297 – 299. Readers may benefit from a brief explanation of "antigenicity" and its purpose in

parasite replication.

Fig1c: '75' missing in the ladder / Fig1d: '150' missing in the ladder

Line 161: it's Fig4d (not 4c)

Results – Fig4: explain a bit more (what you were expecting with GuHCl / mention that GuHCl is a denaturant)

Fig5d: lacks a ladder

In Fig 7a -aTc precedes +aTc but in 7b it's the other way around.

SuppFig5: missing 'C1' in the figure

L265: "prodomian"

Discussion Line 327: "we have shown that PM X is secreted inside the PV (Fig. 6)". Does Fig6 really shows that?

Reviewer #3 (Remarks to the Author):

Thank you for the opportunity to review the manuscript by Mukherjee et al. Therein, the authors present an analysis of the maturation and subcellular activity of plasmepsin X (PM X), an essential protease in malaria parasites for invasion and egress of host cells and a drug target. The manuscript is clearly written and is concise. The study builds on the earlier identification of multiple autoprocessing events within the PM X pro-domain by mapping the position for the semi-pro and mature domains and producing mutants at different cleavage sites within this domain. Elegant approaches were used to conduct this work, However, whilst these cleavages occur after trafficking through the ER, convincingly shown using Brefeldin A, none of them were required for PM X activity or parasite growth in erythrocytes. Therefore, the auto-processing of the N-terminus is not essential.

The N-terminus of the prodomain was shown to target the catalytic domain of PM X to its apical destination(s) and it was also necessary for proteolytic activity of PMX. This provides direct evidence that the pro-domain is needed for trafficking of this protease beyond the endoplasmic reticulum. It also suggests that it is involved in enzyme folding.

The study included high-resolution and high-magnification microscopy platforms to further interrogate the subcellular localization of PM X and several substrates, which have also been attempted in previous studies. The current work builds on this and shows that PMX is partly co-located with AMA1 and EBA175 in micronemes and SUB1 in exonemes. The authors use Western blots to infer that PMX is also secreted into the PV, a valid conclusion. Overall, the study is well executed and uses sophisticated conditional expression systems to unpick difficult answers to the author's questions. Really quite impressive techniques and a paper that I enjoyed reading. I have no critical concerns. Though in my opinion the findings do not add all that much in terms of a major advance on PMX biology or function. This is not a reflection on any of the authors. How this enzyme is regulated and its role with regard to PM IX which has similar substrate specificity but localises differently, still remains unanswered. I reserve any judgement about journal fit and leave this to the Editor(s).

Major feedback.

Figure 4a shows that the prodomain of PMX does not dissociate under acidic conditions. A positive control protein that does dissociate would be very useful to confirm the test conditions.

Figure 8 (especially panel 8) indicates to me that there is only partial co-localization between PMX-

HA, AMA1 and SUB1. I suggest caution in the statement “we observed very strong colocalization” when describing the results. A Pearson’s colocalization index in support of this would be important. Yet the high-resolution imaging and immuno-EM results clearly show only partial colocalization occurred. So, the statement is a little discordant.

Minor feedback.

Figure 1. To assist with ease of reading, it would be helpful to have the different PMX species marked on each of the Western blots and Coomassie gels with a consistent nomenclature also to previous work on these processing events.

Figure 1e. the authors should add to the blot that the antibody used was GFP.

Figure 9, labels of the single and double labelling would be useful.

We thank the reviewers for their constructive comments. Their suggestions have been addressed with further experimentation, explanatory diagrams and extensive manuscript clarifications. Specific points are addressed below.

Reviewer #1 (Remarks to the Author):

PM X processing

1. Line 68, Introduction. Here we are told that PM X 'has a long prodomain (PD) and a C-terminal catalytic domain'. This statement is not referenced, and the rationale for referring to the N-terminal segment as a prodomain is not provided. Given the focus of this manuscript, much better clarification is required here.

Based on the sequence and structural similarity of PM X with other plasmepsins and pepsinogen, two other groups defined the C-terminal catalytic and the N-terminal prodomain in PM X (PMID: 29074775 and 35048450). Importantly, consistent with the earlier work (PMID: 35048450), our mutagenesis work revealed that the L²²¹ is the first residue of the proposed catalytic domain. We have added these relevant citations in the revised version. Additionally, we have rewritten this part of the introduction for better clarity.

2. Lines 95-96 and Fig 1. The 16 kDa recombinant PM X species identified on Coomassie-stained SDS PAGE gels of the purified protein is claimed to represent 'the N-terminal prodomain fragment of rPM X', yet the red arrowed line in Fig 1b only spans the N-terminal one third or so of the prodomain as marked. So presumably the authors are suggesting that the 16 kDa species is not the entire prodomain but a fragment of it. However, this is not actually stated, and the basis of the prodomain/mature domain layout as drawn in Fig 1b is simply not adequately explained. The term 'semi-pro' is then introduced at the end of this paragraph with no explanation of what this actually means. This is careless and confusing, and the authors' conclusions as to how the protein is processed and the basis of the schematic shown are never properly explained. Please re-write this entire section to more accurately state the findings. How do the authors know that the 44 kDa species (which is not marked as such on the gels) is the 'mature' isoform of PM X? Please use such terms appropriately and accurately and in a manner that is supported by the data.

We agree with the reviewer that our original description of the cleavage site identification on recombinant PM X was not easy to follow. Therefore, in the revised version of the manuscript, we have rewritten the entire section. Specifically, we have changed the description of the 16 kDa band as the "N-terminal fragment of the prodomain" (line: 93). The LC/MS experiment performed on the excised band shows the sequence coverage (Supplementary MS file 1). To avoid confusion, we removed terms such as "semi-pro" or "mature" and referred to the different PM X species by their respective molecular weights (p64, p42, etc.).

3. Is it possible that the recombinant proteins are modified by glycosylation, which may prevent more conversion to lower forms? Notably, the p51 rPM X intermediate reported previously by other groups as a rPM X intermediate is completely absent from this protein preparation, the full length being detected only when a PMX inhibitor was present (see PMID: 32109369). Why do the authors not comment on this? Use of the available X-ray structure of PMX (PDB ID: 6ORS) could be used in Fig 1 to better explain the different forms of the enzyme.

Although we did not explicitly test the recombinant PM X for potential glycosylation, a recent paper (PMID: 35048450) on the crystal structure of *Pf*PM X reported the presence of an N-acetylglucosamine molecule. This suggested that the protein was glycosylated during expression. We used the same mammalian HEK293 expression system. As *P. falciparum* blood stage proteins are known to have only rudimentary glycosylation, it is possible that such post translational modification impacted the processing of the recombinant PM X. However, even in the parasites, PM X undergoes partial processing with substantial amount of unprocessed form remaining in the very late stage schizonts.

Although not highlighted in the original manuscript, we detected additional intermediate forms of PM X, of which one runs very close to the reported p51 species. We observed that mutation at the NFLD site abolishes the formation of this band (Fig. 1b-c). Therefore, we believe that cleavage between the F and L within the NFLD tetrapeptide would generate this fragment. The calculated molecular weight based on the amino acid sequence of the fragment is approximately 52 kDa, further supporting our premise. We have highlighted this fragment in the revised manuscript.

4. The Fig 1b legend refers to the marked putative cleavage sites as 'autocatalytic tetrapeptides'. This is an inaccurate way of referring to these sequences, which should instead be referred to as the sequences flanking the putative cleavage sites or scissile bonds (the double slashes at the scissile bond might better be replaced by a downward-pointing arrow or hyphen, and Schechter and Berger nomenclature should be used throughout as conventional). Similarly, the text referring to the sequences for which coverage was obtained by mass spectrometry is poorly written. Please re-write. The red and blue lines with terminal arrow heads refer to the bands in Fig 1a marked with red and blue arrows, but what does the black line in Fig 1b refer to, and how was the SD-IQ cleavage site identified?

We have replaced the nomenclature for the putative cleavage sites in the revised manuscript. As suggested, the cleavage sites are now described as "tetrapeptides flanking the putative cleavage sites". Additionally, we have replaced the double slashes with downward pointing arrows to demonstrate the cleavage sites.

The black line in the original Fig 1b represents the N-terminal fragment from the NFLD mutant form of recombinant PM X (Fig. 1d in the revised manuscript). Because of rearrangement of figures for better clarity, Fig. 1a now shows the schematics, and the earlier black arrow has been replaced with a grey arrow. We performed LC/ MS on this polypeptide following excision from the gel. This revealed the non-tryptic C terminus (Supplementary MS file 3) of the polypeptide. By mutating the sequence flanking the C terminal residue (SDIQ) we further demonstrated that the SDIQ motif is used as an autocatalytic cleavage site (quadruple mutants in Fig 1c-d).

5. Lines 103-112. Here the various mutations are described, and the results presented, but several points are poorly explained. The previously described cleavage specificity of PM X should be described in the Introduction, rather than just referred to obliquely here (lines 107-108). Line 107 also refers to a tetrapeptide motif that appears to be marked by a black line on Fig 1b (but is not mentioned in the figure legend – see point above). Is there any experimental evidence that cleavage actually occurs at this site? This was not clear to this reviewer. The authors must find a much clearer, less ambiguous way of describing their results here. Overall, the results presented in Fig 1 would greatly benefit from an additional schematic (similar to those in Fig 5a) showing the positions in the PM X sequence of the various mutations and how

they altered cleavage. As it stands, it is difficult to assess the effects of the various mutations on PM X processing.

We thank the reviewer for this thoughtful suggestion. We have added a comment regarding the PM X substrate cleavage specificity in the revised manuscript (lines 59-61).

The reviewer's concern about the autocleavage at the SDIQ motif has already been addressed in the point # 4.

We agree with the reviewer that a schematic describing the mutations would be beneficial for the readers. Therefore, we have expanded the data on recombinant PM X with additional schematics demonstrating the details of the mutated constructs used in this study (Fig. 1a and Supplementary Fig. 2). As the C-terminal tag used for parasite expression of PM X was different from that in recombinant PM X, we have added separate schematics for the parasite expressed constructs in Fig. 1e.

6. Regarding the main mutagenesis work, the parasite-expressed tagged PM X second copies would better have been designed to possess a small tag (HA or 3xHA) to prevent artefacts due to the nature of the tag. In the manuscript the authors mentioned the use of a HA-tagged PM X parasite second copy but this construct was not used for the mutagenesis work, a GFP tagged version being used instead. GFP is known to dimerise in solution and can lead to artefactual results. The work presented here should be repeated with the PMX HA tagged version to confirm or not the results obtained with the GFP tagged version and confirm whether or not full length unprocessed PMX is active on native substrates.

As the reviewer rightly pointed, tagging proteins with GFP might lead to artefactual results at times. We chose to tag the second copy PM X with GFP to study the localization of the individual cleavage mutant PM X with respect to SUB1, which was used as a marker for the exonemes. We attempted to tag the C-terminal end of SUB1 with different epitopes, and we were successful only with 3xHA. The PM X-GFP constructs were subsequently introduced into the SUB1-3xHA expressing and the PM X^{apt} transgenic lines.

As expected, the catalytic dead version of PM X-GFP in the PM X^{apt} transgenic line could not support parasite growth and showed no activity *in vitro* against the substrate peptides. The WT-GFP, on the other hand, was correctly targeted, could fully support the parasite growth and showed robust *in vitro* activity (Fig. 3 and Supplementary Fig. 3). These contrasting yet expected phenotypes of the control samples, therefore, ruled out any potential artefact that might result due to the epitope. Moreover, the recombinantly expressed quadruple mutant PM X that had a different epitope tag (8xHis) was also active, further arguing that the activity seen was independent of the artifacts from the tagged GFP. As further epitope tagging of PM X in a series of parasite lines would not be trivial, we prefer not to generate the additional data to reconfirm the same findings.

7. Recombinant and native (parasite) forms of PMX should ideally be compared side by side on SDS-PAGE in order to establish whether the fate of the recombinant protein genuinely mimics that of the parasite proteins. Although the use of protein tags will undoubtedly impact on protein migration within the gel, use of small tags would still allow comparison of the processing profiles. Fig 3 would benefit from an SDS-PAGE showing the PMX pull downs corresponding to the wild type, Quad and D266G mutant.

We thank the reviewer for these thoughtful suggestions. However, as the reviewer pointed out, the recombinant and the native forms of PM X are differently tagged. Consequently, the experiment would require epitope tagging PM X in a series of parasites, which is a great deal of work. The recombinant expression experiments were used mainly to guide our subsequent native PM X experiments. Our mutagenesis assays demonstrated that the cleavage sites are conserved between the recombinant and the parasite-derived PM X. Importantly, for both, we were able to show that the full-length form was active. This strongly supports the main conclusion – autoproteolytic cleavage of PM X is not a prerequisite for its activation.

We performed a western blot following pull down of PM X from the parasites for Fig. 3. The data has now been added as Supplementary Fig. 4.

Role of the prodomain in PM X activity and activation

1. In section 2.3, the authors use CD to detect structural differences between purified PM X at pH 7.5 and pH 5.5 and refer to ‘acidity-induced activation’ of PM X. No secondary structural data are provided to support the suggestion that the CD spectra changes were acidity-induced. CD results should be presented as % composition of secondary structure (alpha, beta, turn). This information is missing in the paper. The authors do not discriminate between activity at low pH and activation by low pH, nor do they prove any causal link between the structural changes they observe and enzyme activity. The authors do not appear to consider the possibility that the complex between the 16 kDa form (which is only a fragment of the prodomain anyway, based on Fig 1 data and the schematic shown in Fig 1b) and the rest of the protein may be non-stoichiometric. If it is, it remains possible that the only a fraction of the purified recombinant PM X that lacks the 16 kDa fragment is enzymatically active. Taken together, their conclusion that the ‘in vitro activity of rPM X is regulated by acid-induced structural changes’ (the subtitle used for Results section 2.3) is a potentially misleading over-interpretation of their data and is not warranted.

In agreement with the reviewer, we performed additional analysis of the CD spectrum data to estimate the secondary structure components for rPM X. We observed a significant decrease in the alpha helical content of rPM X under acidic condition, as compared to pH 7.5 (Table S2).

It is possible that a small fraction of the rPM X lacks the 16 kDa fragment, however when we pull down the N-terminus of PM X, we can clear PM X from the lysate fairly quantitatively (line 94, new Supplementary fig 1). Additionally, the quadruple mutant form, both heterologously expressed and inside the parasites, demonstrated robust substrate cleaving activity under acidic conditions. This suggests that the activity is regulated by the acidity induced conformational changes rather than complete separation of the 16 kDa N terminal fragment. Currently it remains unknown which residues in PM X would be affected due to changes in pH, however, taking references from the recently published works on the crystal structure of *Pf*PM X, we have commented on the potential activation mechanism in the discussion (lines: 296-305).

2. Line 155. after acid-induced ‘activation’ of recombinant PM X, there is still a lot of residual full-length unprocessed form. Why is that? Could glycosylation be present? A recombinant version of the quadruple mutant could also be expressed HEK cells to test its capacity to cleave the Rh2N fluorescent peptide substrate.

The reviewer’s concern about glycosylation of recombinant PM X has been addressed in point # 3.

We thank the reviewer for asking about the quadruple mutant. In the revised manuscript we have provided additional data showing that a quadruple mutant form of recombinant PM X could cleave the Rh2N peptide substrate under acidic pH, and the activity was inhibited by the PM X specific inhibitor CWHM-117 (Fig. 1c-d, right panels; Supplementary Fig. 6). These findings further suggest that the prodomain cleavage is not necessary for PM X activity.

3. Fig4a-c. Dynamic Light Scattering (DLS) or mass photometry techniques should be used to determine the various states of recombinant PM X in solution, and the results added into this figure. This would clarify the presence of a complex between recombinant PMX 44 kDa and the 16 kDa species and indicate whether there is any uncomplexed recombinant PMX 44 kDa form in solution. Peak 1 should be split into several elution fractions as done in the denaturing conditions and the fractions submitted to DLS and SDS-PAGE. Unfortunately, a native gel cannot give a definitive answer as to whether any of the recombinant PMX 44 kDa is in an uncomplexed form.

We thank the reviewer for this useful experiment. As suggested, following size exclusion under acidic pH condition, fractions were collected. DLS and SDS-PAGE analysis showed no difference in the contents of the molecular species that were present in peak subfractions. The measured DLS diameter was that expected for the propiece-containing complex. We have added the datasets as supplementary fig. 5.

Trafficking and function of PM X

1. The immunofluorescence data provided in Fig 5b are unfortunately not compelling in their current form, as the resolution of these images is not sufficiently high to enable discrimination between the various subcellular organelles and the parasite ER (as acknowledged by the authors themselves on lines 241-243). These experiments should be repeated using super-resolution (as in Fig 8) to better define the localisation of the various prodomain mutants. The assertion that the prodomain is required for PM X folding (lines 181-182) is completely speculative and is not addressed by any of these experiments.

Quantification associated with Fig. 5b demonstrated significantly more overlap for the WT and delB mutants (about 80%) than the delA or del PD mutants (30-40%) with the exoneme marker SUB1. The lack of complete overlap between PM X and SUB1 could be due to the additional signal from the free GFP. Importantly, about 90% of the signals for the delA or the del PD mutants were found to be localized in the ER. To address the potential trafficking issue with an orthologous measure, we performed western blots from samples that were collected at the time of egress. As shown in Fig. 5d, we could clearly see discharge of PM X-GFP for the WT and delB samples while that in the delA sample remained completely intracellular. Together with the microscopy in Fig. 5b, the western blot assay could, therefore, convincingly demonstrate the importance of the N-terminal sequence in the trafficking of PM X. We have emphasized the orthologous assessments in the revised text.

We agree with the reviewer that the statement about the prodomain being necessary for PM X folding is speculative, and, therefore, we have removed this sentence from our revised manuscript.

2. The experiment described in Fig 6 simply shows that PM X is not released from exonemes by treatment with the C1. Others have previously shown that exoneme discharge is blocked by this PKG inhibitor, so this is not an unexpected finding. Fig 6 is also the first image in the manuscript to label the various PM X species (P67, P54 and P45). For consistency, similar labelling should be applied throughout where warranted. The pulse-chase experiment shown in Supplementary

Fig 4 does indicate that PM X undergoes some of its processing in a post-ER compartment. However, it shows no more than that. It would have provided much more information to carry out a much longer chase (ideally as a time-course) in order to better visualise conversion of the PM X precursor and obtain data on the kinetics of this. What is the basis of the statement that BFA induces accumulation of newly synthesized parasite proteins in a 'fused ER/Golgi' compartment? This statement is not referenced.

By pulse-chase analysis in presence of brefeldin A, we determined that processing happens in post-ER compartment/s. Further experiment as presented in Fig. 6, demonstrated that like the processed forms, the unprocessed form of PM X gets secreted from the exoemes at the time of egress. Although, we could not precisely localize the cellular site where processing of PM X happens, the data presented in Fig 6 is consistent with the conclusion that the individual cleavage events are independent of each other, generating multiple terminally processed forms of PM X. In this experiment we used the C1 treated samples as negative control for exoeme discharge.

We thank the reviewer for pointing out the labeling issue. We have now labelled all the blots appropriately.

We appreciate the reviewer's concern about the pulse chase experiment. We performed a time course assay to determine the best chase time to get sufficient signal from the processed forms of PM X, following pulsing with radioactive met/cys. Because of the presence of the protein synthesis inhibitor, cycloheximide during the chase, parasites were healthy only up to 60 minutes. Therefore, we had to restrict our data collection to the 60 minute time point.

We have reworded the statement about the fused Golgi/ER. Like higher eukaryotic cells, earlier reports showed that treatment of *P. falciparum* with brefeldin A results in specific accumulation of proteins in the parasite ER. We have included the appropriate references in the revised manuscript (Line: 221).

3. The question over how PM X targets endogenous protein substrates that are themselves localised in different organelles is an interesting one. The experiments presented in Fig 7 were designed to examine this, but are not wholly convincing as they stand, as in Fig 7a SUB1 appears not to undergo processing whether the parasites are aTc-treated or not. Can the authors explain this? The partial co-localisation between the various proteins analysed in Fig 8 is interesting, but does not suggest that PM X comes together with these proteins in a 'precursor compartment'. Whilst it is fine to speculate on this possibility in the Discussion (lines 308-315), no definitive data are provided for this so the suggestion should be removed from the Abstract. The experiments described in Fig 8 and Supplementary Fig 6 should be repeated using schizonts in which egress has been blocked for several hours with C1, to assess whether localisation differences are due to differences in maturity of the schizonts.

As rightly pointed by the reviewer, the SUB1 blot presented in figure 7a was suboptimal and it was difficult for the readers to tell the size difference. We have changed this panel with a better western blot by rerunning the respective lysates.

As the reviewer pointed, based on the partial yet stronger overlap between the microneme localized substrates and PM X we can only speculate about the trafficking of the proteins up to a certain "common" precursor compartment before further segregation. Our Immuno-EM data on

thin section samples further showed the presence of vesicles that are positive for both microneme proteins and PM X as well as single positive vesicles. Further work is needed to understand how the different proteins that are targeted to the different secretory organelles are sorted in the schizonts.

Our original microscopy experiment was done with synchronized schizonts that were pre-treated with C1. As the samples were collected 48-50 hrs. post invasion, the parasites were in the terminal stage of the intracellular cycle. However, in agreement with the reviewer, we have substituted the entire figure panel 8 with new high resolution images taken by confocal airyscan microscopy. We have also performed quantification on these images. Our analysis shows significant colocalization between the microneme proteins AMA1/ EBA-175 and either PM X or SUB1.

We have removed the speculation on a precursor compartment from the abstract.

Other issues:

Fig 1b. The AMA1 processing forms appear to be labelled 'P7' and 'P6' rather than 'P70' and 'P60'. Please correct.

We apologize for the error, which has now been corrected.

Please amend and improve the schematics shown in Fig 5a. Why is delB not aligned with the other mutants? The current layout is sub-optimal and confusing.

We apologize for providing sub-optimal schematics, which has now been modified.

Fig 7. The SUB1 conversion pattern is not clear. Arrows should be used to indicate the p54 and P47 forms.

We apologize for the confusion. In the revised manuscript we have added a better resolved blot for this panel.

Fig 9. Arrows should be used to indicate the type of beads visible on the micrographs. It is presently unclear what the reader is looking at.

We apologize for this exclusion. In the revised manuscript we have added different color arrows to indicate the different beads.

Supplementary Fig 3. The legend here should be improved. Please remind the reader what the plotted samples are.

Modifications have been made as per the suggestion.

Supplementary Fig 6S. IFAs are of poor quality. Inset "regions of interest" are not convincing.

In agreement with the reviewer, we have modified the entire panel and presented newly taken 3D images on samples that were treated with C1 for several hours before processing for IFA. The modified figures are now presented as Fig. 8b in the revised manuscript.

Reviewer #2 (Remarks to the Author):

MAJOR COMMENTS

Fig 1. There is a lot of information in this fig but it takes several readings to work out what is going on and perhaps the figure needs clearer labelling. For example; In 1a, where do the bands not labelled with arrows fit onto the protein map in 1b? The labels on the 1b protein map do not correspond well with the labels on parts c, d and e. With respect to the protein map of 1b (or perhaps show a new map of the GFP fusion protein) where does p90, p77 and p68 fit in 1e?

We agree with the reviewer that the data originally presented in Fig 1 was too complicated to follow without detailed schematic and labeling. Therefore, for better clarity we have modified the schematic for the recombinant PM X data with details about the individual mutation constructs used in this study (Fig. 1a and Supplementary Fig. 2). We have added additional schematics for the parasite expressed second copy PM X and indicated the molecular weights of the different processed forms of PM X (Fig. 1e).

Line 239. Author stated that there is “no overlap of signals” between PM X-3xHA and RON4 / RAP1 signal, but figure shows some degree of overlap. It is a bit odd that Pearson’s coefficients were used to support the conclusions of Fig 5 and not Fig 8. The title of Fig 8 should be changed too because it appears the PMX does partially colocalise with the rhoptry markers but not as much as the microneme markers.

We agree with the reviewer that there was partial overlap between PM X and the rhoptry markers as presented in Fig 8. For better presentation, we have included quantification data obtained from the high resolution confocal ayriscan microscopy. In agreement with the reviewer, we have modified the text associated with Fig 8.

Fig 9. Can the EM panels be labelled with the antibodies used like the fluorescence images? Are we meant to be able to tell the difference between the colloidal gold particles? Perhaps consider indicating the particle size with arrows.

We thank the reviewer for this suggestion. In the revised manuscript we have added arrows with different colors to point the different size beads in Fig 9.

Note, I attempted to view the excel files containing the mass spec data but my computer advised me not to open them because they were corrupted.

We thank the reviewer for pointing out this error. The revised manuscript is now accompanied by corrected supplementary MS files.

MINOR COMMENTS

Line 156. The word “blood stream” is usually written as one word “bloodstream”.

We apologize for the error. It has now been corrected.

Line 265. The word “prodomain” is incorrectly spelled as “prodomian”

We apologize for the error. It has now been corrected.

Line 287. The author may be referring to Supplementary Fig. 2, instead of Supplementary Fig. 3

We apologize for the error. This has now been corrected.

Lines 297 – 299. Readers may benefit from a brief explanation of “antigenicity” and its purpose in parasite replication.

We have modified this sentence in the revised manuscript for better clarity (Lines 316-317 in the revised manuscript).

Fig1c: ‘75’ missing in the ladder / Fig1d: ‘150’ missing in the ladder

We apologize for the error. The revised manuscript now provides corrected labeling.

Line 161: it's Fig4d (not 4c)

We apologize for the error. The revised manuscript now provides corrected labeling.

Results – Fig4: explain a bit more (what you were expecting with GuHCl / mention that GuHCl is a denaturant)

We thank the reviewer for this suggestion and changed the writing as per (Line 170 in the revised manuscript).

Fig5d: lacks a ladder

We apologize for the error. The revised manuscript now provides corrected labeling.

In Fig 7a -aTc precedes +aTc but in 7b it's the other way around.

We apologize for the error, which has now been corrected.

SuppFig5: missing ‘C1’ in the figure

We apologize for the error, which has now been corrected (Supplementary Fig. 9 in the revised manuscript).

L265: “prodomian”

We apologize for the error, which has now been corrected.

Discussion Line 327: “we have shown that PM X is secreted inside the PV (Fig. 6)”. Does Fig6 really shows that?

Previously it was showed that the exoneme discharge takes place just prior to egress releasing SUB1 into the parasitophorous vacuole, where it cleaves multiple substrates including SERA6 that is required for the RBC membrane rupture (PMID: 18083098, 28292906, 22984267). In our assay, while the PKG inhibitor C1 completely blocked PM X secretion, chemical inhibition of SERA6 by E64d could not. This is in line with the SUB1 discharge into the PV, as observed previously (PMID: 29459732). As SERA6 inhibition could not block the RBC poration we were able to detect PM X signal from the culture supernatant.

Reviewer #3 (Remarks to the Author):

Major feedback.

Figure 4a shows that the prodomain of PMX does not dissociate under acidic conditions. A positive control protein that does dissociate would be very useful to confirm the test conditions.

We thank the reviewer for pointing to this valuable control. As pointed out by reviewer # 1, the native gel might not give us a conclusive picture on whether the N-terminal part of the prodomain remains connected with the rest of the protease under acidic conditions. Therefore, we removed the native gel data from our revised manuscript. Instead, we have expanded the size exclusion chromatography experiment, and carried out dynamic light scattering (DLS) on collected fractions under pH 5.5, followed by SDS-PAGE (Supplementary Fig. 5). Importantly we did not observe any obvious difference in the distribution of the difference species of PM X under the tested conditions. As a control we performed size exclusion chromatography on purified human hemoglobin A samples that were preincubated either in pH 7.5 or pH 5.5 buffer. As expected, because of the dissociation of the hemoglobin tetramer under acidic condition, we observed a shift in the elution profile of hemoglobin (Fig. 4a). Altogether, our conclusion that under pH 5.5 the N-terminal segment of the prodomain remains bound to the remaining of PM X stays the same. The new data are added to the revised manuscript.

Figure 8 (especially panel 8) indicates to me that there is only partial co-localization between PMX-HA, AMA1 and SUB1. I suggest caution in the statement “we observed very strong colocalization” when describing the results. A Pearson’s colocalization index in support of this would be important. Yet the high-resolution imaging and immuno-EM results clearly show only partial colocalization occurred. So, the statement is a little discordant.

In agreement with the reviewer, in the revised manuscript we present Pearson’s colocalization index between the different markers in the 3D airyscan images. The data shows partial, yet significantly higher colocalization between the microneme markers and PM X/ SUB1 compared to the rhoptry markers (Fig. 8c). Consistent with the IFAs, immune-EM showed presence of vesicles that contained both PM X and AMA1. It is worth mentioning that we never observed any PM X signal either in the neck or the blub of the rhoptries through immune-EM (data not shown).

Minor feedback.

Figure 1. To assist with ease of reading, it would be helpful to have the different PMX species marked on each of the Western blots and Coomassie gels with a consistent nomenclature also to previous work on these processing events.

We agree with the reviewer that marking of the different processed forms of PM X would be useful for the readers. In the revised manuscript we have the western blots and the Coomassie gels with markers indicating the different PM X species.

Figure 1e. the authors should add to the blot that the antibody used was GFP.

We apologize for the error, which has now been corrected.

Figure 9, labels of the single and double labelling would be useful.

In agreement with the reviewer, in the revised manuscript we have added arrows with different colors to point the different size beads in Fig 9.

Reviewer comments, second round

Reviewer #1 (Remarks to the Author):

The authors have satisfactorily addressed most of the issues raised with the original submission of this manuscript, and the result is a much better written, more accurate and clearer manuscript. A few additional points remain to be addressed to aid clarity and accuracy.

Major comments:

1. Regarding nomenclature of putative cleavage sites, the authors now refer to these as 'tetrapeptides flanking the putative cleavage sites'. This is NOT what was previously suggested by this reviewer. The flanking sequences are not peptides; they are simply flanking sequences, and should be referred to as 'sequences flanking the putative cleavage sites' or 'amino acid residues flanking the cleavage sites'.
2. Please create a space between the top and bottom sections of the western blot in Figure 1e and label both sections to indicate that they are probed with antibodies to GFP and PMV respectively.
3. This reviewer remains unconvinced about the interpretation of the IFA data shown in Figure 5b. Wide-field epifluorescence microscopy does not provide sufficient resolution to make these claims. As previously recommended, these samples should be re-analysed using super-resolution or Airyscan microscopy (as in Figure 8).

Minor comments:

Line 23 (Abstract): please replace 'forms' with 'isoforms'.

Lines 41-43: Please alter this sentence to: 'The disease symptoms result exclusively from the blood stage of infection during which the parasite replicates asexually within host red blood cells (RBCs) by a process termed schizogony'.

Line 101: please remove the word 'tetrapeptide' here and elsewhere.

Lines 218-219 referring to inhibition of organellar secretion by the parasite PKG inhibitor C1. The incorrect reference is cited here; the correct reference to cite is ref 16 (PMID: 23675297).

Line 287: Please replace the phrase '...fails to mature...' with '...fails to undergo any cleavage...'

Lines 463-464: please replace the words '...that abolishes PM X-mediated cleavage...' with '...that is not cleaved by PM X...'

Reviewer #2 (Remarks to the Author):

The authors have satisfactorily addressed most of the reviewers' comments. In particular changes to the diagrams make the complex experiments much easier to follow and it's an enjoyable read.

Reviewer #3 (Remarks to the Author):

The authors have done a great job to address my concerns.

It just remains to indicate in Fig. 8c that the Pearson's correlation corresponds to PMX (left) and SUB1 (right) either on the figure or in the legend.

We thank the reviewers for their comments. We have addressed all of the concerns with additional high-resolution microscopy for Figure 5 and further editing of the manuscript. The edits are highlighted with yellow. Specific points are addressed below.

Reviewer #1 (Remarks to the Author):

The authors have satisfactorily addressed most of the issues raised with the original submission of this manuscript, and the result is a much better written, more accurate and clearer manuscript. A few additional points remain to be addressed to aid clarity and accuracy. Major comments:

1.Regarding nomenclature of putative cleavage sites, the authors now refer to these as 'tetrapeptides flanking the putative cleavage sites'. This is NOT what was previously suggested by this reviewer. The flanking sequences are not peptides; they are simply flanking sequences, and should be referred to as 'sequences flanking the putative cleavage sites' or 'amino acid residues flanking the cleavage sites'.

We have changed the text throughout the manuscript as suggested.

2. Please create a space between the top and bottom sections of the western blot in Figure 1e and label both sections to indicate that they are probed with antibodies to GFP and PMV respectively.

Figure 1e is edited as suggested.

3. This reviewer remains unconvinced about the interpretation of the IFA data shown in Figure 5b. Wide-field epifluorescence microscopy does not provide sufficient resolution to make these claims. As previously recommended, these samples should be re-analysed using super-resolution or Airyscan microscopy (as in Figure 8).

We thank the reviewer for this suggestion. We used confocal Airyscan imaging to determine the localization of the different prodomain mutant forms of PM X. We presented both the 2-dimensional (Figure 5b) and 3-dimensional images (Supplementary Figure 7) for each sample as well as quantification for colocalization with different markers (Figure 5c). Constant with our epifluorescence imaging, Airyscan imaging revealed trafficking defects for the delA and delPD mutant forms of PM X.

Minor comments:

Line 23 (Abstract): please replace 'forms' with 'isoforms'.

This has been changed.

Lines 41-43: Please alter this sentence to: 'The disease symptoms result exclusively from the blood stage of infection during which the parasite replicates asexually within host red blood cells (RBCs) by a process termed schizogony'.

This has been changed.

Line 101: please remove the word 'tetrapeptide' here and elsewhere.

Changes made.

Lines 218-219 referring to inhibition of organellar secretion by the parasite PKG inhibitor C1. The incorrect reference is cited here; the correct reference to cite is ref 16 (PMID: 23675297).

We have corrected this error. The new reference is 19th in the reference list.

Line 287: Please replace the phrase '...fails to mature...' with '...fails to undergo any cleavage...'

This has been changed.

Lines 463-464: please replace the words '...that abolishes PM X-mediated cleavage...' with '...that is not cleaved by PM X...'

This has been changed.

Reviewer #2 (Remarks to the Author):

The authors have satisfactorily addressed most of the reviewers' comments. In particular changes to the diagrams make the complex experiments much easier to follow and it's an enjoyable read.

Reviewer #3 (Remarks to the Author):

The authors have done a great job to address my concerns.

It just remains to indicate in Fig. 8c that the Pearson's correlation corresponds to PMX (left) and SUB1 (right) either on the figure or in the legend.

We have modified the graphs in Fig. 8c, highlighting the proteins (PM X or SUB1) analyzed with the other markers.